# Learning with reinforcement prediction errors in a model of the *Drosophila* mushroom body

James E. M. Bennett [1✉], Andrew Philippides [1] & Thomas Nowotny [1]

Effective decision making in a changing environment demands that accurate predictions are learned about decision outcomes. In *Drosophila*, such learning is orchestrated in part by the mushroom body, where dopamine neurons signal reinforcing stimuli to modulate plasticity presynaptic to mushroom body output neurons. Building on previous mushroom body models, in which dopamine neurons signal absolute reinforcement, we propose instead that dopamine neurons signal reinforcement prediction errors by utilising feedback reinforcement predictions from output neurons. We formulate plasticity rules that minimise prediction errors, verify that output neurons learn accurate reinforcement predictions in simulations, and postulate connectivity that explains more physiological observations than an experimentally constrained model. The constrained and augmented models reproduce a broad range of conditioning and blocking experiments, and we demonstrate that the absence of blocking does not imply the absence of prediction error dependent learning. Our results provide five predictions that can be tested using established experimental methods.

[1] Department of Informatics, University of Sussex, Brighton, UK. ✉email: james.bennett@sussex.ac.uk

Effective decision making benefits from an organism's ability to accurately predict the rewarding and punishing outcomes of each decision, so that it can meaningfully compare the available options and act to bring about the greatest reward. In many scenarios, an organism must learn to associate the valence of each outcome with the sensory cues predicting it. A broadly successful theory of reinforcement learning is the delta rule[1,2], whereby reinforcement predictions (RPs) are updated in proportion to reinforcement prediction errors (RPEs): the difference between predicted and received reinforcements. RPEs are more effective as a learning signal than absolute reinforcement signals because RPEs diminish as the prediction becomes more accurate, adding stability to the learning process. In mammals, RPEs related to rewards are signalled by dopamine neurons (DANs) in the ventral tegmental area and substantia nigra, enabling the brain to implement approximations to the delta rule[3,4]. In *Drosophila melanogaster*, DANs that project to the mushroom body (MB) (Fig. 1a) provide both reward and punishment modulated signals that are required for associative learning[5]. However, to date, MB DAN activity is typically interpreted as signalling absolute reinforcements (either positive or negative) for two reasons: (i) a lack of direct evidence for RPE signals in DANs, and (ii) limited evidence in insects for the blocking phenomenon, in which conditioning of one stimulus can be impaired if it is presented alongside a previously conditioned stimulus, an effect that is indicative of RPE-dependent learning[2,6,7]. Here, we incorporate anatomical and functional data from recent experiments into a computational model of the MB, in which MB DANs do compute RPEs. The model provides a circuit-level description for delta rule learning in the MB, which we use to demonstrate why the absence of blocking does not necessarily imply the absence of RPEs.

The MB is organised into lateral and medial lobes of neuropil in which sensory encoding Kenyon cells (KCs) innervate the dendrites of MB output neurons (MBONs), which modulate behaviour (Fig. 1b). Consistent with its role in associative learning, DAN signals modulate MBON activity via synaptic plasticity at KC → MBON synapses[8–10]. Current models of MB function posit that the MB lobes encode either positive or negative valences of reinforcement signals and actions[10–16]. Most DANs in the protocerebral anterior medial (PAM) cluster (called D$_+$ in the model presented here, Fig. 1c) are activated by rewards, or positive reinforcement (R$_+$), and their activation results in depression at synapses between coactive KCs (K) and MBONs that are thought to induce avoidance behaviours (M$_-$). DANs in the protocerebral posterior lateral 1 (PPL1) cluster (D$_-$) are activated by punishments, i.e. negative reinforcement (R$_-$), and their activation results in depression at synapses between coactive KCs and MBONs that induce approach behaviours (M$_+$). A fly can therefore learn to approach rewarding cues or avoid punishing cues as a result of synaptic depression at KC inputs to avoidance or approach MBONs, respectively.

To date, there is only indirect evidence for RPE signals in MB DANs. DAN activity is modulated by feedforward reinforcement signals, but some DANs also receive excitatory feedback from MBONs[17–20], and it is likely this extends to all MBONs whose axons are proximal to DAN dendrites[21]. We interpret the difference between approach and avoidance MBON firing rates as a RP that motivates behaviour, consistent with the observation that behavioural valence scales with the difference between approach and avoidance MBON firing rates[15]. As such, DANs that integrate feedforward reinforcement signals and feedback RPs from MBONs are primed to signal RPEs for learning. To the best of our knowledge, these latter two features have yet to be incorporated in computational models of the MB[22–24].

Here, we incorporate the experimental data described above to formulate a reduced computational model of the MB circuitry, demonstrate how DANs may compute RPEs, derive a plasticity rule for KC → MBON synapses that minimises RPEs, and verify in simulations that our MB model learns accurate RPs. We identify a limitation to the model that imposes an upper bound on RP magnitudes, and demonstrate how putative connections between DANs, KCs and MBONs[25,26] help circumvent this limitation. Introducing these additional connections yields testable predictions for future experiments as well as explaining a broader range of existing experimental observations that connect DAN and MBON stimulus responses to learning. Lastly, we show that both incarnations of the model—with and without additional connections—capture a wide range of observations from classical conditioning and blocking experiments in *Drosophila*. Different behavioural outcomes in the two models for specific experiments provide further strong experimental predictions.

## Results

**A model of the mushroom body that minimises reinforcement prediction errors.** The MB lobes comprise multiple compartments, each innervated by a different set of MBONs and DANs (Fig. 1b), and each encoding memories for different forms of reinforcement[27], with different longevities[28], and for different stages of memory formation[29]. Nevertheless, compartments appear to contribute to learning by similar mechanisms[9,10,30], and it is reasonable to assume that the process of learning RPs is similar for different forms of reinforcement. We therefore reduce the multicompartmental MB into two compartments, and assign a single, rate-based unit to each class of MBON and DAN (colour-coded in Fig. 1b, c). KCs, however, are modelled as a population, in which each sensory cue selectively activates a unique subset of ten cells. Given that activity in approach and avoidance MBONs—denoted M$_+$ and M$_-$ in our model—respectively bias flies to approach or avoid a cue, $i$, we interpret the difference in their firing rates, $\hat{m}^i = m^i_+ - m^i_-$, as the fly's RP for that cue.

For the purpose of this work, we assume that the MB has only a single objective: to form RPs that are as accurate as possible, i.e. that minimise the RPE. We do this within a multiple-alternative forced choice (MAFC) paradigm (Fig. 1d; also known as a multi-armed bandit) in which a fly is exposed to one or more sensory cues in a given trial, and is forced to choose one. The fly then receives a reinforcement signal, $\hat{r}^i = r^i_+ - r^i_-$, which has both rewarding and punishing components (coming from sources R$_+$ and R$_-$, respectively), and which is specific to the chosen cue. Over several trials, the fly must learn to predict the reinforcements for each cue, and use these predictions to reliably choose the most rewarding cue. We formalise this objective with a cost function that penalises differences between RPs and reinforcements

$$C^{\mathrm{RPE}} = \frac{1}{2}\sum_i \left(\hat{r}^i - \hat{m}^i\right)^2, \qquad (1)$$

where the sum is over all cues, $i$. To minimise $C^{\mathrm{RPE}}$ through learning, we derived a plasticity rule, $\mathcal{P}^{\mathrm{RPE}}$ (full derivation in Methods: Synaptic plasticity):

$$\mathcal{P}^{\mathrm{RPE}}_\pm = \eta\mathbf{k}\left(d_\pm - d_\mp\right). \qquad (2)$$

whereby synaptic weights are updated according to $\mathbf{w}_\pm(t+1) = \mathbf{w}_\pm(t) + \mathcal{P}^{\mathrm{RPE}}_\pm$. Here, $\mathbf{k}$ is a vector of KC firing rates, and we use subscripts '$\pm$' to denote the valence of the neuron: if $+$ ($-$) is considered in $\pm$, then $\mp$ refers to $-$ ($+$), and vice versa. As such, $d_\pm$ refers to the firing rate of either D$_+$ or D$_-$.

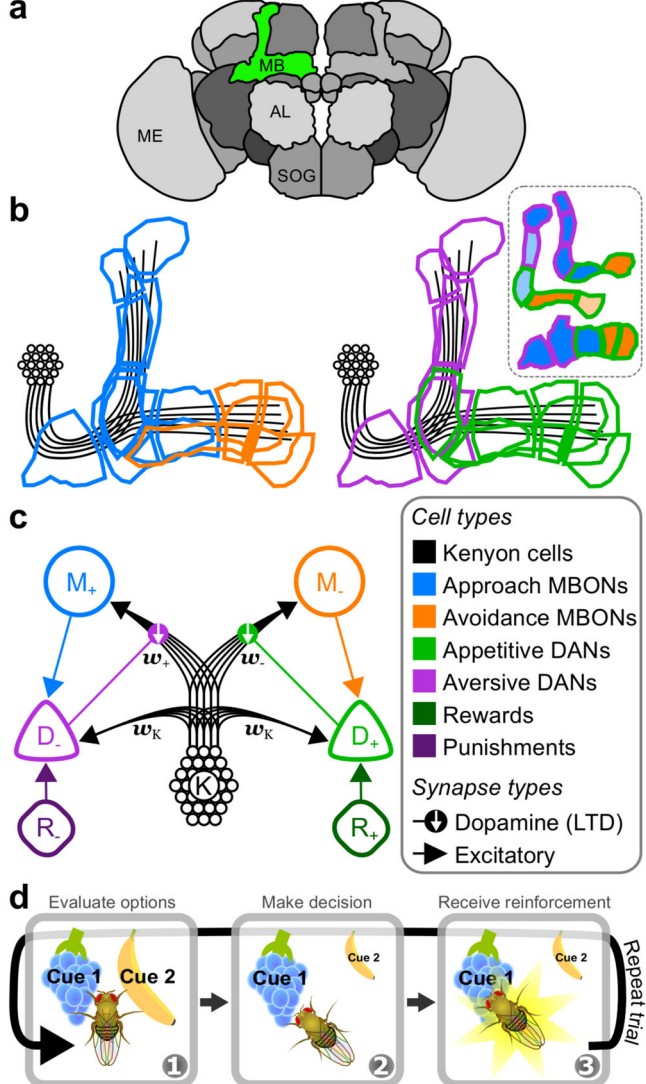

**Fig. 1 Valence-specific model of the mushroom body. a** Schematic of several neuropils that comprise the brain of *Drosophila melanogaster*. The green region highlights the MB in the right hemisphere. Labels: MB mushroom body, AL antennal lobe, SOG suboesophageal ganglion, ME medulla. **b** Outlines of the multiple compartments that tile the lobes of the MB, colour-coded by a broad classification of cell function. Blue: approach MBONs (mushroom body output neurons); orange: avoidance MBONs; purple: aversive DANs (dopamine neurons); green: appetitive DANs; black: KCs. Inset: schematic of the three MB lobes and their compartmentalisation. Top: $\alpha'/\beta'$ lobes; middle: $\alpha/\beta$ lobes; bottom: $\gamma$ lobe. MBON functions (approach or avoidance) are as determined in[15]. Pale colours in the inset correspond to MBONs that exhibit a non-significant bias on behaviour in[15]. **c** Schematic of the VS model. Units are colour-coded according to the cell types in (**b**). KCs connect to MBONs through plastic synapses, and connect to DANs through fixed synapses. Labels: $M_+$—approach MBON; $M_-$, avoidance MBON; $D_-$, aversive DAN, $D_+$, appetitive DAN, K, Kenyon cells, $R_-$, negative reinforcement, $R_+$, positive reinforcement. Lines with arrows: excitatory synapse. Lines with filled circles: synapse releasing dopamine. Downward white arrows: dopamine enhances synaptic long term depression (LTD). **d** Schematic of a single trial for the experimental paradigm in which the model is examined. Panel 1: the model is exposed to some number of cues that are evaluated to yield a cue-specific RP. Panel 2: using the relative RPs, the model makes a decision over which cue to choose. Panel 3: the model receives a reinforcement signal that is associated with the chosen cue, and its RPs for that cue are updated.

The learning rate, $\eta$, must be small (see Methods: Synaptic plasticity) to allow the plasticity rule to average over multiple stimuli as well as stochasticity in the reinforcement schedule (see Methods: Reinforcement schedule). Note that a single DAN, $D_\pm$, only has access to half of the reinforcement and RP information, and by itself does not compute the full RPE. However, the difference between $D_+$ and $D_-$ firing rates does yield the full RPE (see Methods: DAN firing rates):

$$d_+^i - d_-^i = \hat{r}^i - \hat{m}^i. \quad (3)$$

Three features of Eq. (2) are worth highlighting here. First, elevations in $d_\pm$ increase the net amount of synaptic depression at active synapses that impinge on $M_\mp$, which encodes the opposite valence to $D_\pm$, in agreement with experimental data[9,10,30]. Second, the postsynaptic MBON firing rate is not a factor in the plasticity rule, unlike in reinforcement-modulated Hebbian rules[31], yet nevertheless in accordance with experiments[9]. Third, and most problematic, is that Eq. (2) requires synapses to receive dopamine signals from both $D_+$ and $D_-$, conflicting with current experimental findings in which appetitive DANs only modulate plasticity at avoidance MBONs, and similarly for aversive DANs and approach MBONs[8–10,27,32,33]. In what follows, we consider two solutions to this problem. First, we formulate a different cost function to satisfy the valence specificity of the MB anatomy. Second, to avoid shortcomings that arise in the valence-specific model, we propose the existence of additional connectivity in the MB circuit.

**A valence-specific mushroom body model exhibits limited learning**. To accommodate the constraints from experimental data, in which DANs and MBONs of opposite valence are paired in subcompartments of the MB[15,21], we consider an alternative cost function, $C_\pm^{\mathrm{VS}}$, that satisfies this valence specificity:

$$C_\pm^{\mathrm{VS}} = \frac{1}{2}\sum_i \left(r_\mp^i + m_\pm^i\right)^2. \quad (4)$$

We refer to model circuits that adhere to this valence specificity as valence-specific (VS) models. The VS cost function can be minimised by the corresponding VS plasticity rule (see Methods: Synaptic plasticity):

$$\mathcal{P}_\pm^{\mathrm{VS}} = \eta\mathbf{k}\left(\mathbf{w}_K^{\mathrm{T}}\mathbf{k} - d_\mp\right), \quad (5)$$

where $\mathbf{w}_K^{\mathrm{T}}\mathbf{k}$ models the direct excitatory current from KCs to DANs (Methods, Eq. (13)). As required, Eq. (5) maintains the relationship between increased DAN activity and enhanced synaptic depression.

Equation (5) exposes a problem for learning according to our assumed objective in the VS model. The problem arises because $D_\pm$ receives only excitatory inputs. Thus, whenever a cue is present, KC inputs[34] prescribe $D_\pm$ with a minimum, cue-specific firing rate, $d_\pm^i = \mathbf{w}_K^{\mathrm{T}}\mathbf{k}^i + r_\pm^i + m_\mp^i \geq \mathbf{w}_K^{\mathrm{T}}\mathbf{k}^i$. As such, synapses will be depressed ($\mathcal{P}_\mp^{\mathrm{VS}} < 0$) whenever $r_\pm^i + m_\mp^i > 0$. Once $\mathbf{w}_\pm^{\mathrm{T}}\mathbf{k}^i = 0$, the VS model can no longer learn the valence of cue $i$ as synaptic weights cannot become negative. Eventually, RPs for all cues become equal with $\hat{m}^i = 0$, such that choices become random (Supplementary Fig. 1a, b). In this case, $D_+$ and $D_-$ firing rates become equal to the positive and negative reinforcements, respectively, such that the RPE equals the net reinforcement (Supplementary Fig. 1c, d).

A heuristic solution is to add a constant source of potentiation, which acts to restore synaptic weights to a constant, non-zero value. We therefore replace $\mathbf{w}_K^{\mathrm{T}}\mathbf{k}$ in Eq. (5) with a constant, free parameter, $\lambda$:

$$\mathcal{P}_\pm^{\mathrm{VS}\lambda} = \eta\mathbf{k}\left(\lambda - d_\mp\right). \quad (6)$$

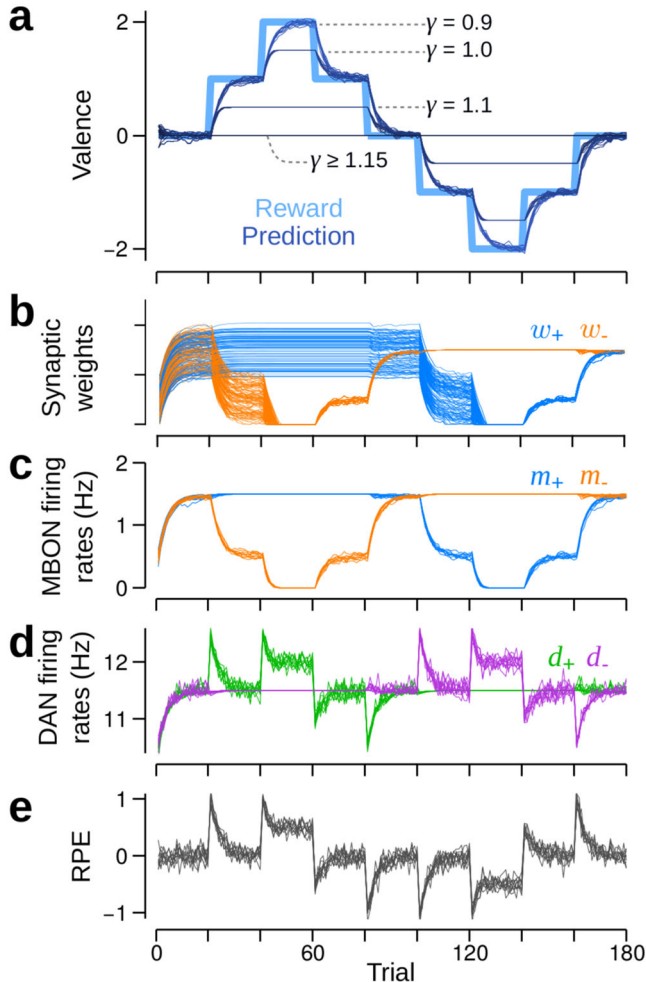

**Fig. 2 RPs in the valence-specific model track reinforcements but only within specified bounds.** Data is shown from 10 runs of the simulation. **a** Reinforcement schedule (excluding the Gaussian white noise; thick, light blue) and RPs (reinforcement predictions; thin, various dark blues). Each shade of dark blue corresponds to simulations using a specific KC (Kenyon cell) → DAN (dopamine neuron) synaptic weight, which is determined by $\gamma$: dark blue through to very dark blue corresponds to $\gamma = 0.9$, 1.0, 1.1, and $\gamma > 1.15$. Dashed lines correspond to $\hat{m}|_{max}$, the theoretical maximum to the RP magnitude, for each value of $\gamma$. **b**–**e** Behaviour of the VS$\lambda$ model when $\gamma = 1.0$. **b** Synaptic weights, $\mathbf{w}_{\pm}$, from KCs to MBONs (mushroom body output neurons). All updated weights undergo the same change, hence their reduced spread after they are all pushed to zero. The remaining spread in weights is due to independent noise that is added to the reinforcement schedules in each of the 10 runs. **c** Firing rates of the $M_+$ (blue) and the $M_-$ (orange) MBONs for 10 runs of the model, with the same reinforcement schedule as in (**a**). MBON firing rates are a good proxy for the mean synaptic weights, as MBONs only receive inputs via those weights. **d** Firing rates for the $D_+$ (green) and the $D_-$ (purple) DANs in response to the reinforcement schedule in (**a**). **e** RPEs (reinforcement prediction errors) given by the difference in firing rates of $D_+$ and $D_-$.

If $\lambda > |r_+ - r_-| + \mathbf{w}_K^T\mathbf{k}$, $\mathcal{P}_{\pm}^{VS\lambda}$ can take both positive and negative values, preventing synaptic weights from being held at zero. This defines a new baseline firing rate for $D_{\pm}$ that is greater than $\mathbf{w}_K^T\mathbf{k}$. Hereafter, we refer to the VS model with plasticity governed by $\mathcal{P}_{\pm}^{VS\lambda}$ as the VS$\lambda$ model.

The VS$\lambda$ model provides only a partial solution, as it is restricted by an upper bound to the magnitude of RPs that can be learned: $|\hat{m}|_{max} = \max(0, \lambda - \mathbf{w}_K^T\mathbf{k})$. This becomes problematic when multiple choices provide reinforcements of the same valence that exceed $|\hat{m}|_{max}$, as the MB will not be able to differentiate their relative values. In addition to increasing $\lambda$, $|\hat{m}|_{max}$ may be increased by reducing KC → DAN synaptic transmission. In Fig. 2a, we set $\mathbf{w}_K = \gamma\mathbf{1}$, with $\mathbf{1}$ a vector of ones, and show RPs for several values of $\gamma$, with $\lambda = 11.5$ (corresponding DAN and MBON firing rates are in Supplementary Fig. 2). The upper bound is reached when $\mathbf{w}_+$ or $\mathbf{w}_-$, and thus the corresponding MBON firing rates, go to zero (an example when $\gamma = 1$ is shown in Fig. 2b, c). These results appear to contradict recent experimental work in which learning was impaired, rather than enhanced, by blocking KC → DAN synaptic transmission[34] (note, the block may have also affected other DAN inputs that impaired learning).

In the VS$\lambda$ model, DAN firing rates begin to exhibit RPE signals. A sudden increase in positive reinforcements, for example at trial 20 in Fig. 2d, results in a sudden increase in $d_+$, which then decays as the excitatory feedback from $M_-$ diminishes as a result of synaptic depression in $\mathbf{w}_-$ (Fig. 2c–e). Similarly, sudden decrements in positive reinforcements, for example at trial 80, are signalled by reductions in $d_+$. However, when the reinforcement magnitude exceeds the upper bound, as in trials 40–60 and 120–140 in Fig. 2, $D_{\pm}$ exhibits sustained elevations in firing rate from baseline by an amount $\max(0, r_{\pm} - |\hat{m}|_{max})$ (Fig. 2d, Supplementary Fig. 2). This constitutes a major prediction from our model.

**A mushroom body circuit with unbounded learning**. In the VS$\lambda$ model, excitatory reinforcement signals can only be partially offset by decrements to $\mathbf{w}_+$ and $\mathbf{w}_-$, resulting in the upper bound to RP magnitudes. To overcome this problem, DANs must receive a source of inhibition. A candidate solution is a circuit in which positive reinforcements, $R_+$, inhibit $D_-$, and similarly, $R_-$ inhibits $D_+$ (illustrated in Fig. 3a). Such inhibitory reinforcement signals have been observed in the $\gamma2$, $\gamma3$, $\gamma4$ and $\gamma5$ compartments of the MB[8,35]. Using the derived plasticity rule, $\mathcal{P}_{\pm}^{VS}$ in Eq. (5), this circuit learns accurate RPs with no upper bound to the RP magnitude (Supplementary Fig. 3b). Hereafter, we refer to the VS model with unbounded learning as the VSu model. Learning is now possible because, when the synaptic weights $\mathbf{w}_{\pm}$ are weak, or when $D_{\mp}$ is inhibited, Eq. (5) specifies that $\mathbf{w}_K^T\mathbf{k} - d_{\mp} > 0$, i.e. synaptic weights will potentiate until the excitatory feedback from $M_{\pm}$ equals the reinforcement-induced feedforward inhibition. Similarly, synapses are depressed in the absence of reinforcement because the excitatory feedback from $M_{\pm}$ to $D_{\mp}$ ensures that $\mathbf{w}_K^T\mathbf{k} - d_{\mp} < 0$ (Supplementary Fig. 3c). Consequently, step changes in reinforcement yield RPE signals in $D_{\mp}$ that always decay to a baseline set by $\mathbf{w}_K^T\mathbf{k}$ (Supplementary Fig. 3d, e). Despite the prevalence in reports of long term synaptic depression in the MB, there exist several lines of evidence for potentiation (or depression of inhibition) as well[10,16,19,36]. However, when reinforcement signals are inhibitory, $D_+$, for example, is excited only by the removal of $R_-$, and not by the appearance of $R_+$ (similarly for $D_-$), counter to the experimental classification of DANs as appetitive (or aversive)[12–14,37].

To ensure that $D_{\pm}$ is also excited by $R_{\pm}$, we could simply add these excitatory inputs to the model. This is unsatisfactory, however, as such inputs would not contribute to learning: they would recapitulate the circuitry of the original VS model, which we have shown cannot learn. We therefore asked whether other variations of the VSu model could learn without an upper bound, and identified three criteria (tabulated in Supplementary Table 1) that must be satisfied to achieve this: (i) learning must be effective, such that positive reinforcement either potentiates excitation of approach behaviours (inhibition of avoidance), or

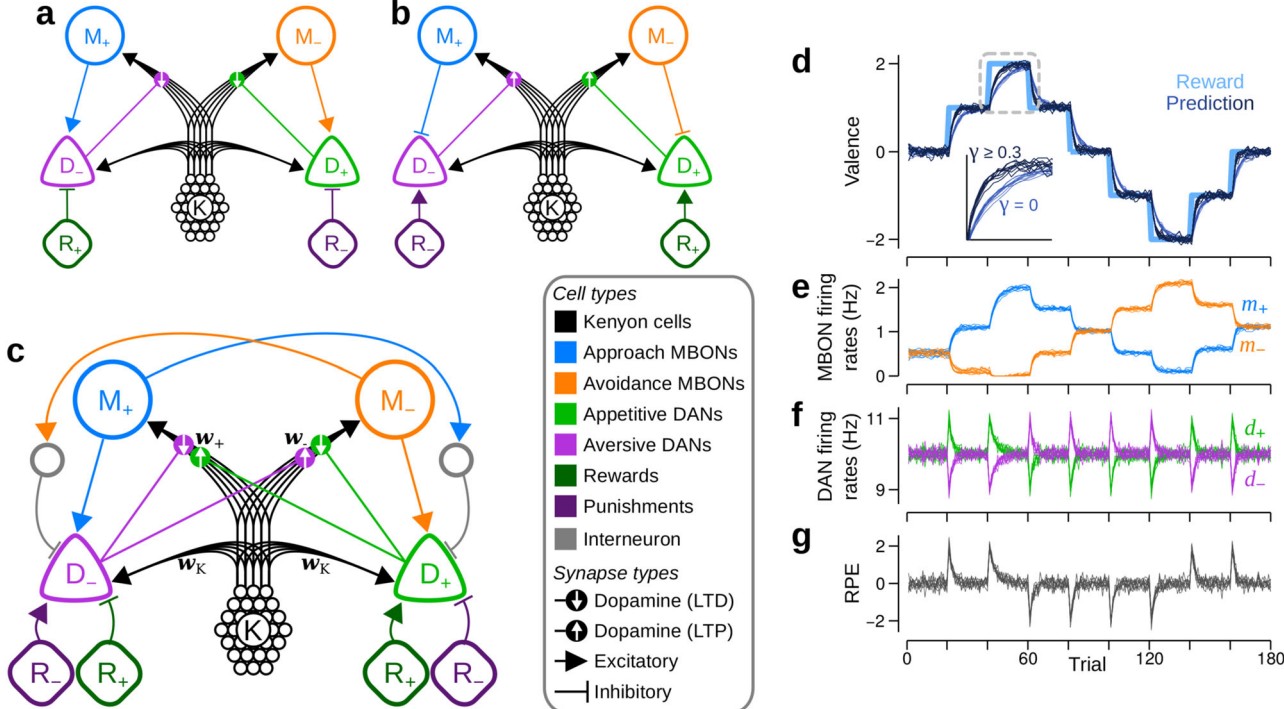

**Fig. 3 Dual versions of unbounded valence-specific (VSu) models can be combined to create the mixed-valence (MV) model, which also learns unbounded RPs. a–c** Schematics of different circuit models. Colours and line styles as in Fig. 1c. **a** One of the dual, VSu models that requires $D_-$ and $D_+$ to be inhibited by positive and negative reinforcements, respectively. Lines with flat ends correspond to inhibitory synapses. **b** The second of the dual VSu models, in which MBONs (mushroom body output neurons) provide inhibitory feedback to DANs of the same valence. Upward arrows in the dopamine synapses denote that dopamine induces long term potentiation (LTP). **c** The MV model, which combines the dual VSu models. Grey units are inhibitory interneurons. **d–g** Each panel exhibits the behaviour from 10 independent runs of the model. **d** RPs are unbounded and accurately track the reinforcements, but the learning speed depends on $\gamma$, the KC (Kenyon cell) → DAN (dopamine neuron) synaptic weights. Thick, light blue: reinforcement schedule; thin, dark blue: $\gamma = 0$; thin, very dark blue: $\gamma \geq 0.3$. Inset: magnified view of the region highlighted by the dashed square, showing how learning is slower when $\gamma = 0$. **e** $M_+$ (blue) and $M_-$ firing rates, respectively $m_+$ and $m_-$, when $\gamma = 1$. **f** $D_+$ and $D_-$ firing rates, respectively $d_+$ and $d_-$, when $\gamma = 1$. **g** RPEs (reinforcement prediction errors) as given by the difference between $D_+$ and $D_-$ firing rates when $\gamma = 1$.

depresses inhibition of approach behaviours (excitation of avoidance), and similarly for negative reinforcement, (ii) learning must be stable, such that excitatory reinforcement signals are offset via learning, either by synaptic depression of feedback excitation, or by potentiation of feedback inhibition, and similarly for inhibitory reinforcement signals, (iii) to be unbounded, learning must involve synaptic potentiation, whether reinforcement signals excite DANs that induce potentiation, or inhibit DANs that induce depression. By following these criteria, we identified a dual version of the VSu circuit in Fig. 3a, which is illustrated in Fig. 3b. In this circuit, $R_+$ excites $D_+$, and $R_-$ excites $D_-$. However, DANs induce synaptic potentiation when activated above baseline, while $M_+$ and $M_-$ are inhibitory, so are interpreted as inducing avoidance and approach behaviours, respectively. Despite their different configurations, RPs are identical in each of the dual MB circuits (Supplementary Fig. 3g–k).

Neither dual model, by itself, captures all of the experimentally established anatomical and physiological properties of the MB. However, by combining them into one (Fig. 3c), we obtain a model that is consistent with the circuit properties observed in experiments, but necessitates additional features that constitute major predictions. First, DANs receive both positive and negative reinforcement signals, which are either excitatory or inhibitory, depending on the valences of the reinforcement and the DAN. Second, in addition to the excitatory feedback from MBONs to DANs of the opposite valence, MBONs also provide feedback to

DANs of the same valence via inhibitory interneurons, which we propose innervate areas targeted by MBON axons and DAN dendrites[21]. We refer to this circuit as the mixed-valence (MV) model, as DANs receive a mixture of both positive and negative valences in both the feedforward reinforcement and feedback RPs, consistent with recent findings in *Drosophila* larvae[26]. Importantly, each DAN in this hybrid model now has access to the full reinforcement signal, $\hat{r}$, and the full RP, $\hat{m}$, or $-\hat{r}$ and $-\hat{m}$, depending on the valence of the DAN. Deriving a plasticity rule (Methods: Synaptic plasticity) to minimise $C_{\pm}^{RPE}$ yields

$$\mathcal{P}_{\pm}^{MV} = \eta \mathbf{k}(\mathbf{w}_K^T \mathbf{k} - d_{\mp}), \qquad (7)$$

which takes the same form as Eq. (5) (except that $d_{\pm}$ depends on more synaptic inputs; see Methods: DAN firing rates), and adheres to our current understanding that plasticity at MBONs is modulated by DANs of the opposite valence. However, Eq. (7) incurs several problems (outlined in Supplementary Discussion), and fails a crucial test: stimulating $D_+$ ($D_-$) as a proxy for reinforcement induces a weak appetitive (aversive) memory only briefly, which then disappears with repeated cue-stimulation pairings (Supplementary Fig. 4), contradicting experiments in which strong, lasting memories are induced by this method[13–15,27,28,32,33,38,39]. One can derive an alternative plasticity rule (Methods: Synaptic plasticity) to minimise $C_{\pm}^{RPE}$, which

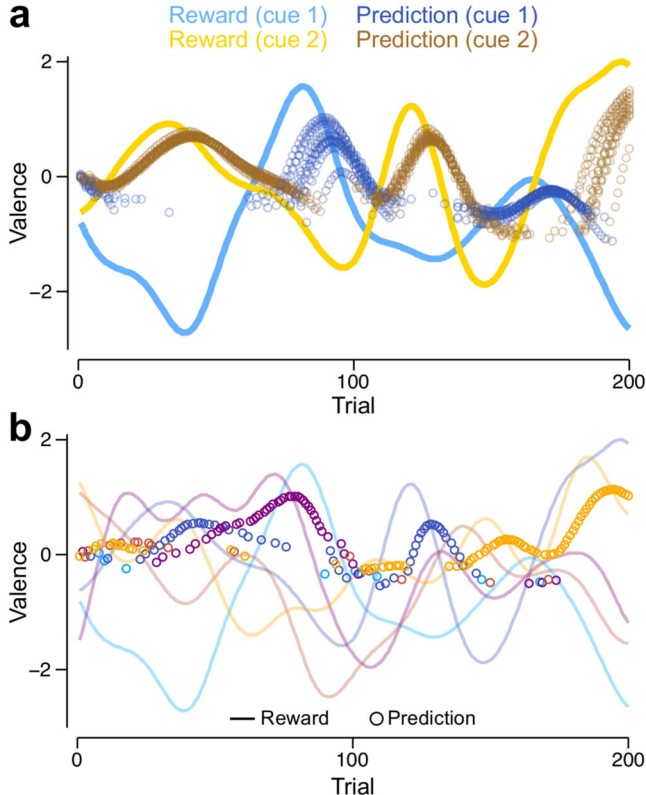

**Fig. 4 Learning RPs in tasks with multiple cues.** RPs are shown for the MV model, but the VSλ model exhibits almost identical behaviour. **a** Reinforcement schedules (lines) and RPs (circles, shown only for the cue chosen on each trial) for two cues (blue: cue 1; yellow: cue 2). RPs are shown for ten independent runs of a simulation using the same reinforcement schedule. **b** Reinforcement schedules (lines) and RPs (circles, shown only for the cue chosen on each trial) for a single run of the model in a task involving 5 cues. Each colour corresponds to reinforcements and predictions for a different cue.

takes a form similar to Eq. (2):

$$\mathcal{P}_{\pm}^{\mathrm{MV}} = \frac{\eta}{2}\mathbf{k}\left(d_{\pm} - d_{\mp}\right). \qquad (8)$$

Although Eq. (8) requires that synapses receive information from DANs of both valences, it does yield strong, lasting memories when $D_{\pm}$ is stimulated as a proxy for reinforcement (Supplementary Fig. 4). We therefore use Eq. (8) for the MV model hereafter, introducing a third major prediction: plasticity at synapses impinging on either approach or avoidance MBONs may be modulated by DANs of both valences.

Figure 3d demonstrates that the MV model accurately tracks changing reinforcements, just as with the dual versions of the VSu model. However, a number of differences from the VSu models can also be seen. First, changing RPs result from changes in the firing rates of both $M_{+}$ and $M_{-}$ (Fig. 3e). Although MBON firing rates show an increasing trend, they eventually stabilise (Supplementary Fig. 5j). Moreover, when $\mathbf{w}_{\pm}$ reach zero, the changes in $\mathbf{w}_{\mp}$ compensate, resulting in larger changes in the firing rate of $M_{\mp}$, as seen between trials 40–60 in Fig. 3e. Second, DANs respond to RPEs, irrespective of the reinforcement's valence: $d_{+}$ and $d_{-}$ increase with positive and negative RPEs, respectively, and decrease with negative and positive RPEs (Fig. 3f, g). Third, blocking KC → DAN synaptic transmission (by setting $\gamma = 0$) slows down learning, but does not abolish it entirely (Fig. 3d). With input from KCs blocked, the baseline firing rate of $D_{\pm}$ is zero, and because any given RPE excites one

DAN type and inhibits the other, only one of either $D_{+}$ or $D_{-}$ can signal the RPE, reducing the magnitude of $d_{\pm} - d_{\mp}$ in Eq. (8), and therefore the speed of learning (Supplementary Fig. 5). To avoid any slowing down to learning, $\mathbf{w}_{K}^{\mathrm{T}}\mathbf{k}$ must be greater than or equal to the RPE. This may explain the 25% reduction in learning performance in experiments that blocked KC → DAN inputs[34], although the block may have also affected other DAN inputs.

**Decision making in a multiple-alternative forced choice task.** We next tested the VSλ and MV models on a task with multiple cues from which to choose. Choices are made using the soft-max function (Eq. (11)), such that the model more reliably chooses one cue over another when cue-specific RPs are more dissimilar. Throughout the task, the cue-specific reinforcements slowly change (see example reinforcement schedules in Fig. 4), and the model must continually update RPs (Fig. 4), according to its plasticity rule, in order to choose the most positively reinforcing cues as possible. Specifically, we update only those synaptic weights that correspond to the chosen cue (see Methods, Eqs. (21, 22)).

In a task with two alternatives, switches in cue choice almost always occur after the actual switch in the reinforcement schedule because of the slow learning rate and the probabilistic nature of decision making (Fig. 4a). The model continues to choose the more rewarding cues when there are as many as 200 (Supplementary Fig. 6a; Fig. 4b shows an example simulation with five cues). Up to ten cues, the trial averaged obtained reinforcement (TAR) becomes more positive with the number of cues (coloured lines in Supplementary Fig. 6a), consistent with the fact that increasing the number of cues increases the maximum TAR for an individual that always selects the most rewarding cue (black solid line, Supplementary Fig. 6a). Increasing the number of cues beyond ten reduces the TAR, which corresponds with choosing the maximally rewarding cue less often (Supplementary Fig. 6b), and a decreasing ability to maintain accurate RPs when synaptic weights are updated for the chosen cue only (Supplementary Fig. 6c; and see Methods: Synaptic plasticity). Despite this latter degradation in performance, the VSλ and MV models are only marginally out-performed by a model with perfect plasticity, whereby RPs for the chosen cue are set to equal the last obtained reinforcement (Supplementary Fig. 6a). Furthermore, when Gaussian white noise is added to the reinforcement schedule, the performance of the perfect plasticity model drops below that of the other models, for which slow learning helps to average over the noise (Supplementary Fig. 6d). The model suffers no noticeable decrement in performance when KC responses to different cues overlap, e.g. when a random 5% of 2000 KCs are assigned to each cue (Supplementary Fig. 6a, e–g).

**Both models capture learned fly behaviours in a variety of conditioning experiments.** To determine how well the VSλ and the MV models capture decision making in flies, we applied them to an experimental paradigm (illustrated in Fig. 5a) in which flies are conditioned to approach or avoid one of two odours. We set λ in the VSλ model to be large enough so as not to limit learning. In each experiment, flies undergo a training stage, during which they are exposed to a conditioned stimulus (CS+) concomitantly with an unconditioned stimulus (US), for example sugar (appetitive training) or electric shock (aversive training). Flies are next exposed to a different stimulus (CS−) without any US. Following training, flies are tested for their behavioural valence with respect to the two odours. The CS+ and CS− are released at opposite ends of a tube. Flies are free to approach or avoid the stimuli by walking towards one end of the tube or the other. In our model,

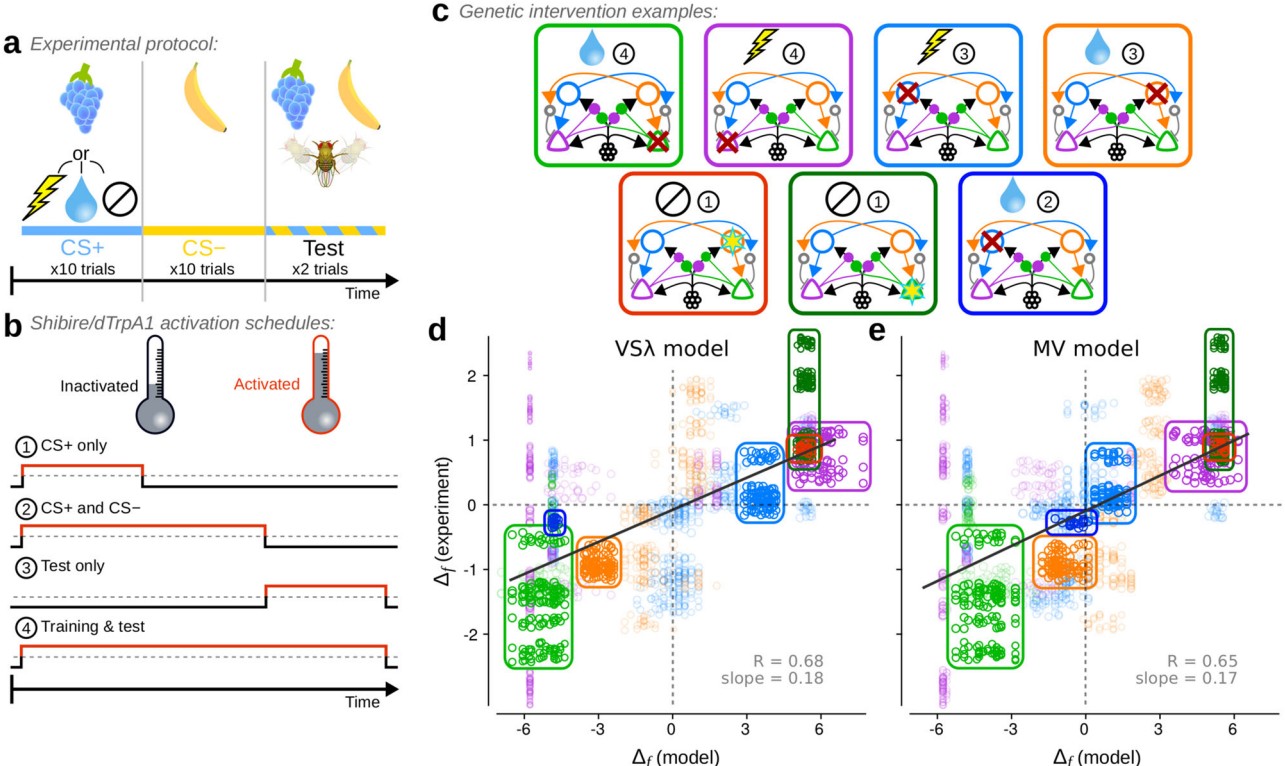

**Fig. 5 Both the modified valence-specific (VSλ) and mixed-valence (MV) models produce choice behaviour that corresponds well with experiments under a broad range of experimental manipulations. a** Schematic of the experimental protocol used to simulate appetitive, aversive and neutral conditioning experiments. **b**, **c** The protocol was extended to simulate genetic interventions used in experiments. **b** Interventions were applied at different stages of a simulation, either (1) during CS+ exposure in training, (2) during CS+ and CS− exposure in training, (3) during testing, or (4) throughout both training and testing. **c** Seven examples of the interventions simulated, each corresponding to encircled data in (**d**, **e**). Red crosses denote interventions that simulate activation of a *shibire* blockade; yellow stars denote interventions that simulate activation of an excitatory current through the dTrpA1 channel. The picture at the top of each panel denotes the reinforcement type, and the encircled number the activation schedule as specified in (**b**). **d** Comparison of $\Delta_f$ measured from the VSλ model and from experiments. **e** Comparison of $\Delta_f$ measured from the MV model and from experiments. Solid grey lines in (**d**, **e**) are weighted least square linear fits with correlation coefficients $R = 0.68$ (0.64, 0.73) and $R = 0.65$ (0.60, 0.69) respectively ($p < 10^{-4}$ for both models using a permutation test; 95% confidence intervals in parentheses using bootstrapping; $n = 92$). Each data point corresponds to a single $\Delta_f$ computed for a batch of 50 simulation runs, and for one pool of experiments using the same intervention from a single study. Dashed grey lines denote $\Delta_f = 0$. The size of each data point scales with its weight in the linear fit. Source data are provided in the Supplementary Data 1 file.

we do not simulate the spatial extent of the tube, nor specific fly actions, but model choice behaviour in a simple manner by applying the softmax function to the current RPs.

In addition to these control experiments, we simulated a variety of interventions frequently used in experiments (Fig. 5a–c). These experiments are determined by four features: (1) US valence (Fig. 5a): appetitive, aversive, or neutral, (2) intervention type (Fig. 5c): inhibition of neuronal output, e.g. by expression of *shibire*, or activation, e.g. by expression of dTrpA1, both of which are controlled by temperature, (3) the intervention schedule (Fig. 5b): during the CS+ only, throughout CS+ and CS−, during test only, or throughout all stages, (4) the target neuron (Fig. 5c): either $M_+$, $M_-$, $D_+$, or $D_-$. Further details of these simulations are provided in Methods: Experimental data and model comparisons.

We compared the models to behavioural results from 439 experiments (including 235 controls), which tested 27 unique combinations of the above four parameters in 14 previous studies[10,13–18,27,28,32,35,36,38,39] (the Source data and experimental details for each experimental intervention used here is provided in Supplementary Data 1). In Fig. 5d, e, we plot a test statistic, $\Delta_f$, that compares behavioural performance indices (PIs) between a specific intervention experiment and its corresponding control, where the PI is +1 if all flies approached the CS+, and −1 if all

flies approached the CS−. When $\Delta_f > 0$, more flies approached the CS+ in the intervention than in the control experiment, and when $\Delta_f < 0$, fewer flies approached the CS+ in the intervention than in the control. Interventions in both models correspond well with those in the experiments: $\Delta_f$ from the VSλ model and experiments are correlated with $R = 0.68$, and $\Delta_f$ from the MV model and experiments are correlated with $R = 0.65$ ($p < 10^{-4}$ for both models). The smaller range in $\Delta_f$ scores from the experimental data are likely a result of the greater difficulty in controlling extraneous variables, resulting in smaller effect sizes.

Four cases of inhibitory interventions exemplify the correspondence of both the VSλ and MV model with experiments, and are highlighted in Fig. 5d, e (light green, purple, blue and orange rings). Also highlighted are two examples of excitatory interventions, in which artificial stimulation of either $D_+$ or $M_-$ during CS+ exposure, without any US, was used to induce an appetitive memory and approach behaviour. The two models yield very similar $\Delta_f$ scores, but not always (Supplementary Fig. 7e). The example highlighted in dark blue in Fig. 5d, e, in which $M_+$ was inhibited throughout appetitive training but not during the test, shows that this intervention had little effect in the MV model, in agreement with experiments[36], but resulted in a strong reduction in the appetitiveness of the CS+ in the VSλ model ($\Delta_f \approx -4.5$). In

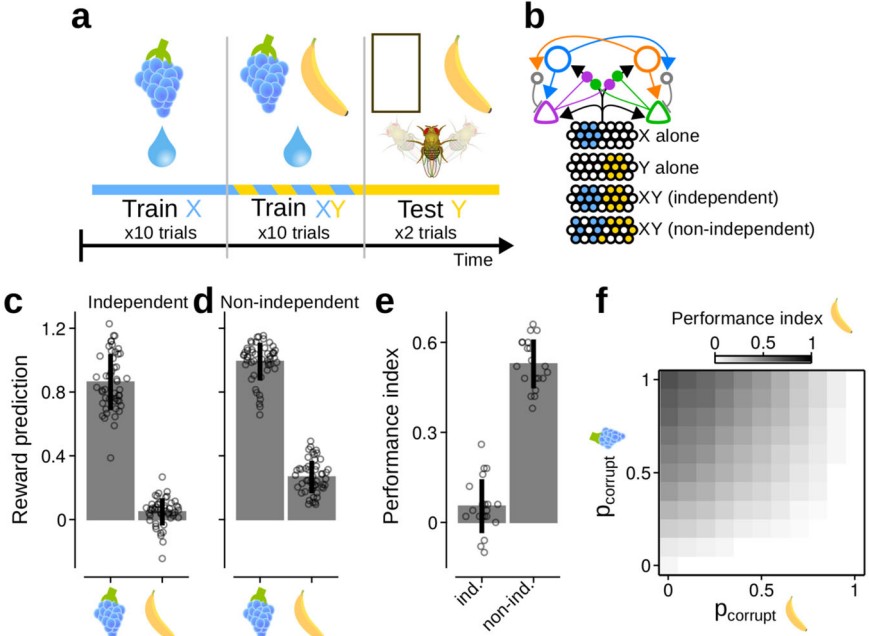

**Fig. 6 The absence of blocking does not imply the absence of reinforcement prediction errors. a** Schematic of the protocol used to simulate blocking experiments. **b** Schematic of KC responses to the two conditioned stimuli, X and Y, when presented alone or as a compound. **c, d** Reinforcement predictions (RPs) for the two stimuli, averaged over the two test trials. Bars and whiskers: mean ± standard deviation. Circles: RPs from individual simulation runs, n = 50. Source data provided in Source data file. **c** Stimuli elicit independent KC responses during compound training. **d** Y gains a positive RP when KC responses to each stimulus are corrupted ($p_{cor}^X = 0.8$, $p_{cor}^Y = 0.2$). **e** Performance indices during the test phase. When stimuli elicit independent KC responses (ind.), the weak RP for Y results in little choice preference for Y over the null option. When KC responses are corrupted (non-ind.), the positive RP for Y results in a relative strong preference for Y over the null option. Bars and whiskers: mean ± standard deviation, n = 20. Circles: PIs from batches of 50 simulation runs. Source data provided in Source data file. **f** Gradations in the blocking effect result from varying degrees of corruptions to X or Y during the compound stimulus training phase.

the Supplementary Note, we analyse the underlying synaptic weight dynamics that lead to this difference in model behaviours. The analyses show that not only does this intervention amplify the difference between CS+ and CS− RPs in the MV model, it also results in faster memory decay in the VSλ model. Hence, the preference for the CS+ is maintained in the MV model, but is diminished in the VSλ model.

The alternative plasticity rule (Eq. (7)) for the MV model yields $\Delta_f$ scores that correspond less well with the experiments (R = 0.55, Supplementary Fig. 7a), in part because associations cannot be induced by pairing a cue with D± stimulation (Supplementary Fig. 4). This conditioning protocol, plus one other (Supplementary Fig. 7c), helps distinguish the two plasticity rules in the MV model, and can be tested experimentally. Lastly, both the VSλ and MV models provide a good fit to re-evaluation experiments[18,19] in which the CS+ or CS− is exposed a second time, without the US, before the test phase (Supplementary Fig. 8, Supplementary Data 2).

**The absence of blocking does not refute the use of reinforcement prediction errors for learning.** When training a subject to associate a compound stimulus, XY, with reinforcement, R, the resulting association between Y and R can be blocked if the subject were previously trained to associate X with R[6,7]. The Rescorla–Wagner model[2] provides an explanation: if X already predicts R during training with XY, there will be no RPE with which to learn associations between Y and R. However, numerous experiments in insects have reported only partial blocking, suggesting that insects may not utilise RPEs for learning[40–43]. This conclusion overlooks a strong assumption in the Rescorla–Wagner model, namely, that neural responses to X and Y are independent.

In the insect MB, KC responses to stimuli X and Y may overlap, and the response to the compound XY does not equal the sum of responses to X and Y[44–46]. Thus, if the MB initially learns that X predicts R, but the ensemble of KCs that respond to X is different to the ensemble that responds to XY, then some of the synapses that encode the learned RP will not be recruited. Consequently, the accuracy of the prediction will be diminished, such that training with XY elicits a RPE and an association between Y and R can be learned. We tested this hypothesis, which constitutes a neural implementation of previous theories[47,48], by simulating the blocking paradigm using the MV model (Fig. 6a).

Two stimuli, X and Y, elicited non-overlapping responses in the KCs (Fig. 6b). When stimuli are encoded independently—that is, the KC response to XY is the sum of responses to X and Y—previously learned X-R associations block the learning of Y-R associations during the XY training phase (Fig. 6c, e), as expected.

To simulate non-independent KC responses during the XY training phase, the KC response to each stimulus was corrupted: some KCs that responded to stimulus X in isolation were silenced, and previously silent KCs were activated (similarly for Y; see Methods: blocking paradigm). This captured, in a controlled manner, non-linear processing that may result, for example, from recurrent inhibition within and upstream of the MB. The average severity of the corruption to stimulus i was determined by $p_{cor}^i$, where $p_{cor}^i = 0.0$ yields no corruption, and $p_{cor}^i = 1.0$ yields full corruption. Corrupting the KC response to X allows a weak Y-R association to be learned (Fig. 6d), which translates into a behavioural preference for Y during the test (Fig. 6e). Varying the degree of corruption to stimulus X and Y results in variable degrees of blocking (Fig. 6f). The blocking effect was maximal when $p_{cor}^X = 0$, and absent when $p_{cor}^X = 1$. However, even in the

absence of blocking, corruption to Y during compound training prevents learned associations being carried over to the test phase, giving the appearance of blocking. These results provide a unifying framework with which to understand inconsistencies between blocking experiments in insects. Importantly, the variability in blocking can be explained without refuting the RPE hypothesis.

## Discussion

**Overview**. Successful decision making relies on the ability to accurately predict, and thus reliably compare, the outcomes of choices that are available to an agent. The delta rule, as developed by Rescorla and Wagner[2], updates beliefs in proportion to a prediction error, providing a method to learn accurate and stable predictions. In this work, we have investigated the hypothesis that, in *Drosophila melanogaster*, the MB implements the delta rule. We posit that approach and avoidance MBONs together encode RPs, and that feedback from MBONs to DANs, if subtracted from feedforward reinforcement signals, endows DANs with the ability to compute RPEs, which are used to modulate synaptic plasticity. We formulated a plasticity rule that minimises RPEs, and verified the effectiveness of the rule in simulations of MAFC tasks. We demonstrated how the established valence-specific circuitry of the MB restricted the learned RPs to within a given range, and postulated cross-compartmental connections, from MBONs to DANs, that could overcome this restriction. Such cross-compartmental connections are found in *Drosophila* larvae, but their functional relevance is unknown[25,26]. We have thus presented two MB models that yield RPEs in DAN activity and that learn accurate RPs: (i) the VS$\lambda$ model, in which plasticity incorporates a constant source of synaptic potentiation; (ii) the MV model, in which we propose mixed-valence connectivity between DANs, MBONs and KC → MBON synapses. Both the VS$\lambda$ and the MV models receive equally good support from behavioural experiments in which different genetic interventions impaired learning, while the MV model provides a mechanistic account for a greater variety of physiological changes that occur in individual neurons after learning. It is plausible, and can be beneficial, for both the VS$\lambda$ and MV models to operate in parallel in the MB, as separately learning positive and negative aspects of decision outcomes, if they arise from independent sources, is important for context-dependent modulation of behaviour. Such learning has been proposed for the mammalian basal ganglia[49]. We have also demonstrated why the absence of strong blocking effects in insect experiments does not necessarily imply that insects do not utilise RPEs for learning.

**Predictions**. The models yield predictions that can be tested using established experimental protocols. Below, we specify which model supports each prediction.

*Prediction 1—both models*. Responses in single DANs to the unconditioned stimulus (US), when paired with a CS+, should decay towards a baseline over successive CS ± US pairings, as a result of the learned changes in MBON firing rates. To the best of our knowledge, only one study has measured DAN responses throughout several CS–US pairings in *Drosophila*[50]. Consistent with DAN responses in our model, Dylla et al.[50] reported such decaying responses in DANs in the γ- and β′-lobes during paired CS+ and US stimulation. However, they reported similar decaying responses when the CS+ and US were unpaired (separated by 90 s) that were not significantly different from the paired condition. The authors concluded that DANs do not exhibit RPEs, and that the decaying DAN responses were a

result of non-associative plasticity. An alternative interpretation is that a 90 s gap between CS+ and US does not induce DAN responses that are significantly different from the paired condition, and that additional processes prevent the behavioural expression of learning. Ultimately, the evidence for either effect is insufficient. Furthermore, Dylla et al. observed increased CS+ responses in DANs after training. Conversely, after training in our models—i.e. when the US was set to zero—DAN responses to the CS+ decreased. Interpreting post-training activity in DANs as responses to the CS+ alone, or alternatively as responses to an omitted US, are equally valid in our model because the CS+ and US always occurred together. Resolving time within trials in our models would allow us to better address this conflict with experiments. The Dylla et al. results are, however, consistent with the temporal difference (TD) learning rule[51,52] (as are studies on second order conditioning in *Drosophila*[53,54]), of which the Rescorla–Wagner rule used in our work is a simplified case. We discuss this further in the Supplementary Discussion, as well as features of the TD learning rule, and experimental factors, which may explain why the expected changes in DAN responses to the CS and US were not observed in previous studies[12,37].

*Prediction 2—VS$\lambda$ model*. After repeated CS ± US pairings, a sufficiently large reinforcement will prevent the DAN firing rate from decaying back to its baseline response to the CS+ in isolation. Here, sufficiently large means that the inequality required for learning accurate RPs, $\lambda > |r_+ - r_-| + \mathbf{w}_K^T \mathbf{k}$, is not satisfied. Because KC → DAN input, $\mathbf{w}_K^T \mathbf{k}$, may be difficult to isolate in experiments, sufficiency could be guaranteed by ensuring the reinforcement satisfies $|r_+ - r_-| > \lambda$. That is, pairing a CS + with a novel reward (punishment) that would more than double the stabilised $D_+$ ($D_-$) firing rate, where the stabilised firing rate is achieved after repeated exposure to the CS+ in isolation. Note that, if $\lambda$ were to adapt to the reinforcement magnitude, this would be a difficult prediction to falsify.

*Prediction 3—MV model*. The valence of a DAN is defined by its response to RPEs, rather than to reinforcements per se. Thus, DANs previously thought to be excited by positive (negative) reinforcement are in fact excited by positive (negative) RPEs. For example, a reduction in electric shock magnitude, after an initial period of training, would elicit an excitatory (inhibitory) response in appetitive (aversive) DANs. Felsenberg et al.[18,19] provide indirect evidence for this. The authors trained flies on a CS+, then re-exposed the fly to the CS+ without the US. For an appetitive (aversive) US, CS+ re-exposure would have yielded a negative (positive) RPE. By blocking synaptic transmission from aversive (appetitive) DANs during CS+ re-exposure, the authors prevented the extinction of learned approach (avoidance). Such responses are consistent with those of mammalian midbrain DANs, which are excited (inhibited) by unexpected appetitive (aversive) reinforcements[3,55–57].

*Prediction 4—both models*. In the MV model, learning is mediated by simultaneous plasticity at both approach and avoidance MBON inputs. The converse, that plasticity at approach and avoidance MBONs is independent, would support the VS$\lambda$ model. Appetitive conditioning does indeed potentiate responses in MB-V3/α3 and MVP2/γ1-pedc approach MBONs[16,36], and depress responses in M4β′/β′2mp and M6/γ5β′2a avoidance MBONs[10]. Similarly, removal of an expected aversive stimulus, which constitutes a positive RPE, depresses M6/γ5β′2a avoidance MBONs[19]. In addition, aversive conditioning depresses responses in MPV2/γ1-pedc and MB-V2/α2α′2 approach MBONs[9,30], and

potentiates responses in M4$\beta'$/$\beta'$2mp and M6/$\gamma$5$\beta'$2a avoidance MBONs[10,29]. However, the potentiation of M4$\beta'$ and M6 MBONs is at least partially a result of depressed feedforward inhibition from the MVP2 MBON[16,19]. To the best of our knowledge, simultaneous changes in approach and avoidance MBON activity has not yet been observed. A consequence of this coordinated plasticity is that, if plasticity onto one MBON type is blocked (e.g. the synaptic weights cannot be depressed any further), plasticity at the other MBON type should compensate.

*Prediction 5—MV model.* DANs of both valence modulate plasticity at MBONs of a single valence. This is a result of using the plasticity rule specified by Eq. (8), which better explains the experimental data than Eq. (7) (Fig. 5d, e, Supplementary Fig. 7a). In contrast, anatomical and functional experimental data suggest that, in each MB compartment, the DANs and MBONs have opposite valences[21,58]. However, the GAL4 lines used to label DANs in the PAM cluster often include as many as 20–30 cells each, and it has not yet been determined whether all labelled DANs exhibit the same valence preference. Similarly, the valence encoded by MBONs is not always obvious. In[15], for example, it is not clear whether optogenetically activated MBONs biased flies to approach the light stimulus, or to exhibit no-go behaviour that kept them within the light. In larval *Drosophila*, there are several examples of cross-compartmental DANs and MBONs[25,59], but a full account of the valence encoded by these neurons is yet to be provided. In adult *Drosophila*, $\gamma$1-pedc MBONs deliver cross-compartmental inhibition, such that M4/6 MBONs are effectively modulated by both aversive PPL1-$\gamma$1-pedc DANs and appetitive PAM DANs[16,19].

**Other models of learning in the mushroom body.** We are not the first to present a MB model that makes effective decisions after learning about multiple reinforced cues[22–24]. However, these models utilise absolute reinforcement signals, as well as bounded synapses that cannot strengthen indefinitely with continued reinforcements. Thus, given enough training, these models would not differentiate between two cues that were associated with reinforcements of the same sign, but different magnitudes. Carefully designed mechanisms are therefore required to promote stability as well as differentiability of same sign, different magnitude reinforcements. Our model builds upon these studies by incorporating feedback from MBONs to DANs, which allows KC → MBON synapses to accurately encode the reinforcement magnitude and sign with stable fixed points that are reached when the RPE signalled by DANs decays to zero. Alternative mechanisms that may promote stability and differentiability are forgetting[60] (e.g. by synaptic weight decay), or adaptation in DAN responses[61]. Exploring these possibilities in a MB model for comparison with the RPE hypothesis is well worth while, but goes beyond the scope of this work.

**Model limitations.** Central to this work is the assumption that the MB has only a single objective: to minimise the RPE. In reality, an organism must satisfy multiple objectives that may be mutually opposed. In *Drosophila*, anatomically segregated DANs in the $\gamma$-lobe encode water rewards, sugar rewards, and motor activity[8,13,14,27], suggesting that *Drosophila* do indeed learn to satisfy multiple objectives. Multi-objective optimisation is a challenging problem, and goes beyond the scope of this work. Nevertheless, for many objectives, the principle that accurate predictions aid decision making, which forms the basis of this work, still applies.

For simplicity, our simulations compress all events within a trial to a single point in time, and are therefore unable to address

some time-dependent features of learning. For example, activating DANs either before or after cue exposure can induce memories with opposite valences[28,62,63]; in locusts, the relative timing of KC and MBON spikes is important[64,65], though not necessarily in *Drosophila*[9]. Nor have we addressed the credit assignment problem: how to associate a cue with reinforcement when they do not occur simultaneously. A candidate solution is TD learning[51,52], whereby reinforcement information is back-propagated in time to all cues that predict it. While DAN responses in the MB hint at TD learning[50], it is not yet clear how the MB circuitry could implement it. An alternative solution is an eligibility trace[52,66], which enables synaptic weights to be updated upon reinforcement even after presynaptic activity has ceased.

Lastly, our work here addresses memory acquisition, but not memory consolidation, which is supported by distinct circuits within the MB[67]. Incorporating memory stabilising mechanisms may help to better align our simulations of genetic interventions with fly behaviour in conditioning experiments.

**Blocking experiments.** By incorporating the fact that KC responses to compound stimuli are non-linear combinations of their responses to the components[44–46], we used our model to demonstrate why the lack of evidence for blocking in insects[40–43] cannot be taken as evidence against RPE-dependent learning in insects. Our model provides a neural circuit instantiation of similar arguments in the literature, whereby variable degrees of blocking can be explained if the brain utilises representations of stimulus configurations, or latent causes, which allow learned associations to be generalised between a compound stimulus and its individual elements by varying amounts[47,48,68,69]. The effects of such configural representations on blocking are more likely when the component stimuli are similar, for example, if they engage the same sensory modality, as was the case in[40–43]. By using component stimuli that do engage different sensory modalities, experiments with locusts have indeed uncovered strong blocking effects[70].

**Summary.** We have developed a model of the MB that goes beyond previous models by incorporating feedback from MBONs to DANs, and shown how such a MB circuit can learn accurate RPs through DAN mediated RPE signals. The model provides a basis for understanding a broad range of behavioural experiments, and reveals limitations to learning given the anatomical data currently available from the MB. Those limitations may be overcome with additional connectivity between DANs, MBONs and KCs, which provide five strong predictions from our work.

## Methods

**Experimental paradigm.** In all but the last two results sections, we apply our model to a multi-armed bandit paradigm[52,71] comprising a sequence of trials, in which the model is forced to choose between a number of cues, each cue being associated with its own reinforcement schedule. In each trial, the reinforcement signal may have either positive valence (reward) or negative valence (punishment), which changes over trials. Initially, the fly is naive to the cue-specific reinforcements. Thus, in order to reliably choose the most rewarding cue, it must learn, over successive trials, to accurately predict the reinforcements for each cue. Individual trials comprise three stages in the following order (illustrated in Fig. 1d): (i) the model is exposed to and computes RPs for all cues, (ii) a choice probability is assigned to each cue using a softmax function (described below), with the largest probability assigned to the cue that predicts the most positive reinforcement, (iii) a single cue is chosen probabilistically, according to the choice probabilities, and the model receives reinforcement with magnitude $r_+$ (positive reinforcement, or reward) or $r_-$ (negative reinforcement, or punishment). The fly uses this reinforcement signal to update its cue-specific RP.

## Simulations

*Connectivity and synaptic weights.* KC → MBON: KCs (K in Fig. 1c) constitute the sensory inputs (described below) in our models. Sensory information is transmitted from the KCs, of which there are $N_K$, to two MBONs, $M_+$ and $M_-$, through excitatory, feedforward synapses. For simplicity, we use a subscript '$+$' to label

positive valence (e.g. reward or approach) and '−' to label negative valence (e.g. punishment or avoidance). $K_i$ synapses onto $M_\pm$ with a synaptic weight $w_{\pm i}$, which is initialised with $w_{\pm i} = 0.1\xi_{\pm i}$ for each run of the model, where $\xi_{\pm i}$ is a uniform random variable in the range 0–1.

KC → DAN: KCs drive excitatory responses in DANs from the PPL1 cluster[34]. In our model, we assume that KCs also provide input to appetitive DANs in the PAM cluster. Thus, $K_i$ drives $D_\pm$ through unmodifiable, excitatory synapses with weights, $\mathbf{w}_K = \gamma\mathbf{1}$, where $\mathbf{1} = [1, 1, \ldots, 1]^T$ is a vector of ones of length $N_K$.

MBON → DAN: MBONs provide excitatory feedback to their respective DANs[17–19]. In both the valence-specific (VS) and mixed-valence (MV) models, $M_\pm$ synapses onto $D_\mp$ with unit synaptic weight. In the mixed-valence (MV) model, $M_\pm$ also provides inhibitory feedback to $D_\pm$ via an inhibitory interneuron, but we do not model the interneuron explicitly. Thus, we describe the feedback weight simply as $w_M = 1$, and specify whether the input is excitatory or inhibitory in the firing rate equation for $D_\pm$ (Eqs. (13) and (14)).

*Inputs and KC sensory representation.* Projection neurons from the antennal lobe and optic lobes provide a substantial majority of inputs to KCs in the MB. These inputs carry olfactory and visual information and, together with recurrent inhibition from the anterior paired lateral neuron, drive a sparse representation of sensory information in ~5–10% of the KCs[72–74]. For simplicity, we bypass the computations performed in nuclei upstream of the KCs, and assign a unique population of 10 KCs to each cue. Thus, for $N_c$ cues, we simulate $N_K = 10N_c$ KCs. Each KC is always activated by its assigned cue, and each active KC, $j$, is given the same firing rate, $k_j = 1$ Hz. In a subset of simulations used for Supplementary Fig. 6a, c–e, we simulate 2000 KCs, where each KC is assigned to a cue with probability $p = 0.05$, so that 5% of KCs, on average, are active for a given cue. In these simulations, we normalised the total KC firing rates for each cue, $i$, such that $\sum_j k_j^i = 10$ Hz. This ensured that the multiplicative effect of KC firing rates on the speed of learning (Eqs. (2) and (5)) does not confound the interpretation of our results.

*MBON firing rates and reinforcement predictions.* Neurons are modelled as linear–non-linear (LN) units that output a firing rate, $y$, equal to the rectified linear sum of their inputs, $\mathbf{x}$:

$$y = f\left(\sum_j w_j x_j\right) \tag{9}$$

where $f(z) = \max(0, z)$ is the rectifying nonlinearity. Equation (9) can be written more concisely in vector notation: $y = f(\mathbf{w}^T\mathbf{x})$, where $\mathbf{w}^T = [w_1, \ldots, w_N]$ for $N$ presynaptic neurons, and superscript T denotes the transpose. Throughout this text, bold fonts denote vectors.

At the beginning of each trial, MBON firing rates, and thus RPs, are computed for each cue. The firing rate, $m_\pm$, of MBON $M_\pm$, signals the amount of positive (or negative) reinforcement associated with a given cue, labelled $i$, according to

$$m_\pm^i = f\left(\mathbf{w}_\pm^T \mathbf{k}^i\right), \tag{10}$$

where $\mathbf{k}^i$ is the vector of KC responses to stimulus $i$, and $\mathbf{w}_\pm$ are plastic, excitatory synaptic weights. The net reinforcement predicted by sensory cue $i$ is then determined by $\hat{m}^i = m_+^i - m_-^i$.

*Decision making.* In each trial, RPs for all cues are compared, and the model is forced to decide which cue should be chosen. Decisions are made probabilistically using a softmax function, $p(i)$, which specifies the probability of choosing cue $i$ as a function of the differences between its RP and the RPs of every other cue:

$$\begin{aligned} p(i) &= \frac{\exp(\beta\hat{m}^i)}{\sum_j \exp(\beta\hat{m}^j)} \\ &= \left(1 + \sum_{j\neq i} \exp\left(\beta(\hat{m}^j - \hat{m}^i)\right)\right)^{-1}, \end{aligned} \tag{11}$$

where $\beta$ is a constant (analogous to the inverse temperature in thermodynamics) and modulates the extent to which $p(i)$ increases or decreases with respect to $\hat{m}^j - \hat{m}^i$. When $\beta = 0$, choices are independent of the learned valence, and each of the $M$ available options are chosen with equal probability, $p(i) = M^{-1}$. When $\beta = \infty$, decisions are made deterministically, such that the cue with the most positive RP is always chosen. For the MAFC task, the cue that is ultimately chosen on a given trial is determined by drawing a single, random sample, $\xi$, from a uniform distribution in the range 0–1, and selecting a cue, $q$, such that

$$q = \max\left\{x \in \mathbb{N} \mid \sum_{i=1}^x p(i) \leq \xi, \ 1 \leq x \leq M\right\}. \tag{12}$$

*DAN firing rates.* Once a cue has been chosen, the RP specific to that cue is fed back to the DANs where they are compared against the actual reinforcement, $\hat{r}^i = r_+^i - r_-^i$, received in that trial, where $r_\pm$ is the magnitude of reinforcement signal $R_\pm$. Given the chosen cue, $q$, $D_\pm$ firing rates in the VS models are given by

$$d_\pm^q = f\left(r_\pm^q + w_M m_\mp^q + \mathbf{w}_K^T\mathbf{k}^q\right), \tag{13}$$

whereas, in the MV model, $D_\pm$ is given by

$$d_\pm^q = f\left(r_\pm^q - r_\mp^q - w_M\left(m_\pm^q - m_\mp^q\right) + \mathbf{w}_K^T\mathbf{k}^q\right). \tag{14}$$

We set $w_M = 1$, such that the difference in DAN firing rates yields the RPE for cue $q$:

$$\begin{aligned} \hat{d}^q &= d_+^q - d_-^q \\ &= \begin{cases} \hat{r}^q - \hat{m}^q, & \text{for VS models} \\ 2(\hat{r}^q - \hat{m}^q), & \text{for MV model} \end{cases} \end{aligned} \tag{15}$$

where $\hat{d}^q$ for the MV model is valid when $\mathbf{w}_K^T\mathbf{k}^q > |\hat{r}^q - \hat{m}^q|$. When the inequality is not satisfied, the precise expression for $\hat{d}^q$ in the MV model, taking into consideration the non-linear rectification in $d_+$ and $d_-$, is

$$\hat{d}^q = (\hat{r}^q - \hat{m}^q) + \text{sgn}(\hat{r}^q - \hat{m}^q)\min\left(|\hat{r}^q - \hat{m}^q|, \mathbf{w}_K^T\mathbf{k}^q\right). \tag{16}$$

**Synaptic plasticity.** We assume that the objective of the MB is to form accurate RPs, which minimise RPEs. This objective can be formulated as

$$\begin{aligned} C^{\text{RPE}} &= \frac{1}{2}\sum_i \left(\hat{r}^i - \hat{m}^i\right)^2 \\ &= \frac{1}{2}\sum_i \left(r_+^i - r_-^i - \left(\mathbf{w}_+^T\mathbf{k}^i - \mathbf{w}_-^T\mathbf{k}^i\right)\right)^2, \end{aligned} \tag{17}$$

where the sum is over all cues, $i$, $\mathbf{w}_+^T\mathbf{k}^i$ is the firing rate of $M_+$, expressed as the weighted input from the KC population response, $\mathbf{k}^i$, through synapses with strength $\mathbf{w}_+$, and similarly for $\mathbf{w}_-^T\mathbf{k}^i$. Learning an accurate RP amounts to minimising $C^{\text{RPE}}$ by modifying the synaptic weights. Assuming that inputs onto approach and avoidance MBONs are modified independently[9], we perform gradient descent on $C^{\text{RPE}}$ with respect to $\mathbf{w}_+$ and $\mathbf{w}_-$ separately. The plasticity rule, $\mathcal{P}_\pm^{\text{RPE}}$, is then defined by the negative gradient:

$$\begin{aligned} \mathcal{P}_\pm^{\text{RPE}} &= -\eta\frac{\partial C^{\text{RPE}}}{\partial\mathbf{w}_\pm} \\ &\approx \eta\mathbf{k}\left(r_\pm - r_\mp - \left(\mathbf{w}_\pm^T\mathbf{k} - \mathbf{w}_\mp^T\mathbf{k}\right)\right) \\ &= \eta\mathbf{k}(d_\pm - d_\mp), \end{aligned} \tag{18}$$

where $\eta$ is the learning rate, and the last line is reached by substituting in the DAN firing rates, $d_\pm$, for the VS model. If instead $d_\pm$ is used from the MV model, it is possible to write plasticity rules that minimise $C^{\text{RPE}}$ in two ways (respectively Eq. (7) and Eq. (8) in Results), either:

$$\mathcal{P}_\pm^{\text{MV}} = \eta\mathbf{k}\left(\mathbf{w}_K\mathbf{k} - d_\mp\right), \text{ or}$$
$$\mathcal{P}_\pm^{\text{MV}} = \frac{\eta}{2}\mathbf{k}(d_\pm - d_\mp),$$

where the factor 1/2 in Eq. (8) accommodates the factor of 2 in Eq. (15). The two equations are equivalent when DAN firing rates are not clipped by rectification, but behave differently when the rates are rectified (Supplementary Fig. 4). We use Eq. (8) throughout the main text, and compare model behaviours for both Eqs. (8) and (7) in Supplementary Fig. 7.

We take a similar approach to derive the VS plasticity rule, but use a valence-specific cost function

$$\begin{aligned} C_\pm^{\text{VS}} &= \frac{1}{2}\sum_i \left(r_\mp^i + m_\pm^i\right)^2 \\ &= \frac{1}{2}\sum_i \left(r_\mp^i + \mathbf{w}_\pm^T\mathbf{k}^i\right)^2. \end{aligned} \tag{19}$$

We derive the plasticity rule, $\mathcal{P}_\pm^{\text{VS}}$, by gradient descent on $C_\pm^{\text{VS}}$:

$$\begin{aligned} \mathcal{P}_\pm^{\text{VS}} &= -\eta\frac{\partial C^{\text{VS}}}{\partial\mathbf{w}_\pm} \\ &\approx -\eta\mathbf{k}(r_\mp + \mathbf{w}_\pm^T\mathbf{k}) \\ &= \eta\mathbf{k}(\mathbf{w}_K^T\mathbf{k} - d_\mp), \end{aligned} \tag{20}$$

where $d_\pm$ are computed according to the VS model. These plasticity rules are in fact only an approximation to gradient descent, and hold true only when: (i) the DAN firing rates are not clipped by the non-linear rectification; (ii) the learning rate, $\eta$, is sufficiently small, which allows us to dispense of the sum over cues, assuming instead that plasticity minimises a running average of the cost. Here, sufficiently small means that $\eta < \left(2\sum_j k_j^i\right)^{-1}$ for all cues, $i$, which ensures that learning does not result in unstable oscillations in RPs. The plasticity rule therefore describes the mean drift in synaptic weights over several trials. This need not be at odds with rapid learning in insects, as small synaptic weight changes may yield large behavioural changes in our model, depending on the softmax parameter $\beta$ in Eq. (11). For Figs. 2, 3, we use $\eta = 2.5 \times 10^{-2}$; for Fig. 4, we use $\eta = 10^{-1}$; for Fig. 5, we use $\eta = 5 \times 10^{-2}$. We set $\eta \to \eta/2$ for the MV model, because each DAN in the MV model encodes the full RPE, as opposed to half the RPE in the VS model. This ensures that synaptic weight updates have the same magnitude for a given RPE in both models. In the simulations, we use Eqs. (19) and (20) to specify discrete updates to the synaptic weights at the end of each trial, $t$, conditioned on the

chosen cue, $q$. Specifically, the update for the VS model is given by

$$
\begin{aligned}
\mathbf{w}_\pm(t+1) &= \mathbf{w}_\pm(t) + \mathcal{P}_\pm^{VS} \\
&= \mathbf{w}_\pm(t) + \eta \mathbf{k}^q(t)\Big(\mathbf{w}_K^T(t)\mathbf{k}^q(t) - d_\mp^q(t)\Big),
\end{aligned} \tag{21}
$$

and for the MV model by

$$
\begin{aligned}
\mathbf{w}_\pm(t+1) &= \mathbf{w}_\pm(t) + \mathcal{P}_\pm^{MV} \\
&= \mathbf{w}_\pm(t) + \eta \mathbf{k}^q(t)\Big(d_\pm^q(t) - d_\mp^q(t)\Big).
\end{aligned} \tag{22}
$$

where the superscript $q$ specifies the firing rate of each neuron in the presence of cue $q$ alone, under the assumption that this cue dominates the neural activity at the point of receiving its corresponding reinforcement signal. The update equation for the VS model with the modified plasticity rule (which we call the VS$\lambda$ model) is

$$
\begin{aligned}
\mathbf{w}_\pm(t+1) &= \mathbf{w}_\pm(t) + \mathcal{P}_\pm^{VS\lambda} \\
&= \mathbf{w}_\pm(t) + \eta \mathbf{k}^q(t)\Big(\lambda - d_\mp^q(t)\Big).
\end{aligned} \tag{23}
$$

Note that the plasticity rule is not a function of the postsynaptic MBON firing rate (except indirectly through the DAN firing rate). This is possible because a separate plasticity rule exists for synapses impinging on each MBON, negating the need to label the postsynaptic neuron via its firing rate, as would be the case in three-factor Hebbian rules that are typically used in models of reinforcement-modulated learning[31].

**Reinforcement schedule**. At the end of each trial, a reinforcement signal specific to sensory cue $i$ is provided. Reinforcements, $r^i$, take continuous values, and are drawn on each trial, $t$, from a normal distribution, $r^i(t) \sim \mathcal{N}(\mu_i(t), \sigma_R)$, with mean $\mu_i(t)$ and standard deviation $\sigma_R$. The reinforcement signals that arrive at DANs, $R_+$ and $R_-$ in Fig. 1d, have amplitudes $r_+^i = \max(0, r^i)$ and $r_-^i = \min(0, r^i)$, respectively. Over the course of a simulation run, $\mu_i(t)$ is varied according to a predetermined schedule, and $\sigma_R$ is fixed. Thus, at different stages throughout each experiment, the most rewarding cue may switch between the multiple alternatives. Unless otherwise stated, $\sigma_R = 0.1$. The reinforcement schedules were as follows. For Figs. 2, 3, and Supplementary Figs. 1–3, 5, $\mu_1(t=1) = 0$, and was held fixed for 20 trials, then underwent a step change of $+1$ at trials 21, 41, 141, and 161, and a step change of $-1$ at trials 61, 81, 101, and 121. For Fig. 4 and Supplementary Fig. 6, $\mu_i(t) = Ag(\xi_\mu(t)) + \sigma_R\xi_\sigma(t)$, where $\xi_\mu(t)$ and $\xi_\sigma(t)$ are Gaussian white noise processes with zero mean and unit variance, such that $\xi_\mu$ determines the mean reinforcement, and $\xi_\sigma(t)$ determines the additive noise on trial $t$. A low pass filter, $g(\xi_\mu) = F^{-1}\{F\{\xi_\mu\}F\{G(0,\tau)\}\}$, is applied to $\xi_\mu$, where $G(0,\tau)$ is a Gaussian function with unit area, centred on 0, and with standard deviation $\tau = 10$ trials, $F\{\cdot\}$ is the Fourier transform, and $F^{-1}\{\cdot\}$ is the inverse Fourier transform. Because the Fourier transform method of filtering assumes $\xi_\mu(1) = \xi_\mu(N_t + 1)$, where $N_t$ is the number of trials, we generate $\xi_\mu$ for 250 trials, then delete the first 50 trials after filtering. Finally, the reinforcement amplitude is determined by $A = 2/\max_t(g(\xi_\mu(t)))$.

**Experimental data and model comparisons**. The VS$\lambda$ and MV models were compared to experimental data by simulating an often used conditioning protocol. To align with experiments, each simulation utilised the following procedure (Fig. 5a): (i) in the first stage of training, the model is exposed to a single cue by itself, the CS+, for ten trials, with reinforcements drawn from a normal distribution, $\mathcal{N}(\mu, 0.1)$, where $\mu$ was chosen according to whether appetitive ($\mu = 1$), aversive ($\mu = -1$), or neutral ($\mu = 0$) conditioning was simulated, (ii) during the next 10 trials, the model is exposed to a second cue by itself, the CS−, with reinforcements drawn from a distribution with $\mu = 0$ and the same variance as for the CS+, (iii) the final two trials comprise the test stage, in which the model is exposed to both cue 1 and cue 2, as in the MAFC task with two alternatives, with $\mu = 0$ for both cues. On each test trial, the model is forced to choose either cue 1 or cue 2, using Eq. (12). We used 10 trials per training stage as, given the parameters for $\eta$ (learning rate) and $\beta$ (inverse temperature), it took this many trials for the mean performance (see below for how performance is measured) across multiple runs of the simulation to plateau at, or near, the maximum possible value. The test was run for only two trials as synaptic plasticity was allowed to continue during the test stage, under the assumption that the formation of new CS+ related short term memories[18,19] might alter the behaviour of flies in the test stage of experiments.

For each simulation, we applied one of many possible additional protocol features, in which neuronal activity was manipulated. We therefore define a protocol as a unique combination of four features:

1. US valence (Fig. 5a): (i) appetitive ($\mu = 1$), (ii) aversive ($\mu = -1$), (iii) neutral ($\mu = 0$). To ensure the VS$\lambda$ model was not limited in learning RPs as large as $\pm 1$, we set $\lambda = 12$.
2. Intervention type (Fig. 5c), which modified the target neuron's output firing rate from $y_{targ}$ to $\bar{y}_{targ}$: (i) block of neuronal output (e.g. by *shibire*), which was simulated by multiplicatively scaling the manipulated neuron's firing rate, such that $\bar{y}_{targ} = 0.1y_{targ}$, (ii) neuronal activation (e.g. by dTrpA1),

which was simulated by adding a constant current, such that $\bar{y}_{targ} = y_{targ} + 5$.
3. The intervention type was applied following one of four activation schedules (Fig. 5b): (i) during the CS+ only, (ii) throughout training (CS+ and CS−), (iii) during test only, (iv) throughout all stages.
4. The target neuron to which the intervention type was applied (Fig. 5c): (i) $M_+$, (ii) $M_-$, (iii) $D_+$, (iv) or $D_-$.

We compared behavioural data from experiments with that of our model for 27 of the 96 possible variations of these four features. These data were obtained from 14 published studies[10,13–18,27,28,32,35,36,38,39], comprised of 439 experiments that followed conditioning protocols similar to that used in our simulations (235 controls with no intervention, 204 experiments with one of the 27 interventions).

Simulations were run in batches of 50, each batch yielding 100 choices from the two test trials. From these choices, we computed a performance index (PI$_{mod}$) given by

$$
PI_{mod} = \left(\frac{n_+ - n_-}{n_+ + n_-}\right), \tag{24}
$$

where $n_+$ is the number of choices for the CS+ and $n_-$ for the CS−. A distribution of PIs for each protocol was obtained by running 20 such batches. PIs from the experimental data were extracted by eye from the 14 published papers. These PIs are computed in a similar way as for the model, but where $n_+$ and $n_-$ correspond to the number of flies that approached the CS+ or CS−, respectively. We averaged across PIs from experiments that used the same intervention in the same study, reducing the number of intervention samples from 204 to 92, against which PIs from the simulations were compared.

To measure the effect strength of each intervention in both the model and the experiments, we converted PIs into fractions of flies (or model runs) that chose the CS+, $f = (PI + 1)/2$, then computed a test statistic, $\Delta_f$, which compares $f_c$ from control to $f_i$ from intervention experiments, given that the underlying data is binomially distributed, as follows:

$$
\Delta_f = \frac{f_i - f_c}{\sqrt{\frac{1}{N_{fly}}(f_i + f_c)\left(1 - \frac{1}{2}(f_i + f_c)\right)}}, \tag{25}
$$

where $N_{fly}$ is the number of flies used in that experiment. The binomial distribution adjustment to $f_i - f_c$ accounts for the bounded nature of $f$ between 0 and 1. As such, for a given absolute difference, $f_i - f_c$, $\Delta_f$ is larger when $f_c$ is near to 1 than when it is near to 0.5. That is, small changes to excellent memory performance imply a stronger effect than small changes to mediocre memory performance. Because $N_{fly}$ was rarely stated in the studies we assessed, we set $N_{fly} = 50$, which is typical for experiments of this nature, and corresponds to the number of runs in each batch of simulations from which a single PI was computed from the model.

To examine the correspondence between PIs from the model and experiments, we fit a weighted linear regression to the experimental versus model $\Delta_f$ data using the MATLAB R2012a function *robustfit*, which computes iteratively reweighted least square fits with a bisquare weighting function. We then computed the Pearson correlation coefficient, $R$, of the weighted data using the weights, $\mathbf{w}_r$, provided by *robustfit*, according to

$$
R = \frac{\text{cov}\left(\mathbf{w}_r\Delta_f^{mod}, \mathbf{w}_r\Delta_f^{exp}\right)}{\sigma_{mod}\sigma_{exp}}, \tag{26}
$$

where $\sigma_{mod}$ and $\sigma_{exp}$ are the standard deviations of $\mathbf{w}_r\Delta_f^{mod}$ and $\mathbf{w}_r\Delta_f^{exp}$, respectively, and bold fonts denote vectors for all data points in either the model or experimental data sets. We determined the probability with which $R$ comes from a distribution with zero mean by reshuffling the weighted data.

**Blocking paradigm**. Blocking experiments were simulated by pairing a CS, X, with rewards drawn from a Gaussian distribution, $\mathcal{N}(1, 0.1)$ for 10 trials, followed by 10 trials in which a compound stimulus, XY, was paired with rewards drawn from the same distribution. After conditioning, a test phase comprised two trials in which the two available options were Y or null, whereby the null option elicited a RP equal to zero. Rewards drawn from $\mathcal{N}(0, 0.1)$ were provided in each test trial. Performance indices (PIs) were computed in the same way as for the comparison between models and experimental data, using 20 batches of 50 simulation runs, yielding 20 PIs. Here, however, $n_+$ denotes the number of choices for cue Y and $n_-$ for the null option. The two stimuli, X and Y, were represented by responses in two, non-overlapping subsets of 20 KCs each. When either stimulus was presented alone (X during the first conditioning phase, Y during the test phase), 10 KCs in each subset were activated. During the compound training phase, each stimulus, $i$, was independently corrupted by silencing each active KC with a probability $p_{cor}^i$. For each KC silenced, a previously silent KC was activated, but only within the sub-population corresponding to that stimulus, thus ensuring that both stimuli remained non-overlapping. The KC responses to each individual stimulus were then added for the compound XY stimulus.

**Reporting summary**. Further information on research design is available in the Nature Research Reporting Summary linked to this article.

## Data availability

All experimental data in Fig. 5 and Supplementary Figs. 7, 8 were lifted from figures in the cited publications. No additional experimental data was generated in this work[75]. Source data are provided with this paper.

## Code availability

All of the code that was used for running simulations and analysing data are made available on the archived github repository https://doi.org/10.5281/zenodo.453142[075]. The most recent version of this code can be found at: https://github.com/BrainsOnBoard/paper_RPEs_in_drosophila_mb.

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

## Acknowledgements

Special thanks to Eleni Vasilaki for helpful discussions and feedback on the mathematical formulations, James Marshal for feedback on the paper, and the Waddell and Vogels labs for fruitful discussions on learning in *Drosophila*. Thanks also to the members of the Brains on Board team for their critical feedback at various points throughout this project. This work was funded by the EPSRC (Brains on Board project, grant number EP/P006094/1).

## Author contributions

J.E.M.B. conceived the model, wrote the code, generated and analysed the data. J.E.M.B. and T.N. conceived the reinforcement schedules and ideal agents to test the models. J.E.M.B., A.P. and T.N. wrote and revised the paper.

## Competing interests

The authors declare no competing interests.
