## [Peer Review File · Nature Communications]

Reviewer #1 (Remarks to the Author):

The authors describe a computational model intended to simulate neuronal circuits underlying associative learning in *Drosophila melanogaster*. It is well-established that the neuronal circuitry mediating associative learning is localized to the mushroom body of the *Drosophila* brain. This circuit is subject of intense research and serves as a general model system for analyzing principles of neuronal coding and plasticity. Therefore, the overall topic is certainly of interest not only for specialists working on *Drosophila*, but also for a more general scientific readership.

Here, it is proposed that computing prediction errors, a phenomenon well-known for the dopaminergic system of the mammalian brain, might also underlie associative learning in the mushroom body of the *Drosophila* brain. The idea that minimizing prediction errors through learning is not only present in mammalian brains but also in insects is not new and intriguing (see, for example, Hammer, 1997, *TINS* 20, No. 6, 1997). This reviewer is not in the position to judge the mathematical details of the computational model itself. However, I am most comfortable with the details of the circuitry and the behavioral aspects. I like the overall approach taken by the authors and the concept of integrating positive reinforcement (reward) with negative reinforcement (punishment) and feedback loops. However, I would like to raise several points of concern the authors might want to consider.

First, experimental evidence for prediction error coding in *Drosophila* is lacking. A major argument for prediction error coding in mammals is the phenomenon of “blocking”. In this case, learning does not take place if the learned outcome is already predicted by a sensory cue. In insects, several researchers have searched for this phenomenon, but without any success. In honey bees, blocking has been reported, but later on this finding has not been confirmed. Therefore, from a behavioral point of view it is not clear whether any prediction error is actually computed and needed. The authors should provide at least arguments why learning through prediction errors must be inevitably present. As it stands, I do not see any reason to assume that prediction error coding in the insect brain is actually present.

Second, the authors postulate a feedback from MBONs back to DANs. Currently, there is enough evidence that this feedback is actually present (e.g., Ichinose et al., *Elife* 2015, 4:e10719). One does not have to postulate it. It has already been postulated by Riemensperger et al. (2005), cited by the authors. However, recent connectomics studies have determined massive connections between KCs and DANs (reciprocal synapses) (see also Cervantes-Sandoval et al., *Elife* 2017, 6: e23789) and, moreover, massive direct feedforward connections between DANs and MBONs (the EM-connectomics studies cited). These connections do not appear in the authors’ models. I recommend that the authors base their models on the data from the most recent connectomics studies.

Minor remarks:

1. Abstract: the authors write “...and use simulations to verify that MBONs learn accurate predictions.” This is an overstatement. A simulation can only postulate this. Verification must be done experimentally.
2. The authors often write about reward. In *Drosophila*, experiments are often done using punishment as reinforcement. Perhaps, the authors should verbalize very early in the introduction that in their models punishment reflects negative reward.
3. Introduction, line 63: The authors write “To the best of our knowledge, these latter two features have yet to be incorporated in computational models of the MB).” Okay, but why do these two

features HAVE to be integrated if there is no evidence that reward prediction error coding exists in insects?

4. Introduction, line 72: Feedback from MBONs onto DANs is nowadays well-described. Given that the MB connectome of the larval MB and parts of the adult MB is available one can assume that all possible connections are known. When the first connectomics data were published the community was astonished about the feedforward connection from DANs onto MBONs. Do the authors see any possibility to make sense of that?

5. Results part: This part is nicely explained. In fact, I really like how the authors describe their models and the rationale behind it. And the figures provide very good illustrations.

6. Discussion, line 369: the authors write “to the best of our knowledge, only one study has measured DAN responses throughout learning in *Drosophila*”. This is factually wrong. Please have a closer look at Mao and Davis, *Front Neural Circuits*. 2009, 3:5, and Riemensperger et al., (2005) (cited). In the latter publication, it was explicitly searched for prediction error coding, but with a negative outcome. I think these data should not be ignored.

Overall, I recommend that the authors incorporate these ideas in an updated version of the manuscript. The comments are not intended to unduly criticize the study but to help to improve it. However, I do not see any reason to believe that prediction errors exist in insects, and there are data that actually speak in favor that they do not.

Reviewer #2 (Remarks to the Author):

Bennett et al develop several potential computational models of the *Drosophila* mushroom body positing that it computes reward prediction error. While there has been substantial interest in the field in the general idea of the mushroom body as computing prediction error, no one has fleshed out the idea in detail, certainly not to the point of developing computational models that generate specific experimental predictions. By doing so, this manuscript makes a substantial contribution to the field. Therefore, I support publication in principle - but there are some points where the models should be clarified, better motivated, or indeed perhaps altered.

Conceptual points:

My biggest confusion is that I couldn't follow the reasoning leading up to the development of the MV model. It seems that the model in Fig 3a solves the bounded learning problem in the VSlambda model and the only experimental difficulty is that real DANs are excited, not just inhibited, by rewards/punishments. Why not just solve this by adding only both $r+$ and $r-$ as inputs to DANs, rather than adding all the extra things in Fig 3c which haven't been observed experimentally? so then $d- = r- - r+ + w+*k + wk*k$

I assume there is some reason why this doesn't work, but this is never explained (or if it was, I didn't understand it).

Similarly, it seems to me that the original RPE cost function and plasticity rule:

$$P(\text{RPE})+ = \eta * k * (r+ - r- - (w+*k - w-*k))$$

could be satisfied without requiring that $D-$ has to potentiate $M-$ as well as depressing $M+$ (the problematic Prediction 5). Couldn't you only suppose that $D-$ is excited by $R-$, inhibited by $R+$, excited

by M^+ and inhibited by M^- ? that is: $d^- = r^- - r^+ + w^+ * k - w^- * k + w_k * k$

If this is impossible or fails some other criterion, I would like to see why.

In general, the MV model is a lot more complicated than the VSlambda model so I would like to understand what benefit each new complication brings.

Prediction 1

line 381 - Dylla et al observe increased response to CS+ in DANs. But this is not consistent with either the VSlambda or the MV model.

VSlambda: $d^- = r^- + w^+ * k + w_k * k$

→ w^+ should go down with training, so d^- response to CS+ should also go down with training

MV model: $d^- = r^- - r^+ + w^+ * k - w^- * k + w_k * k$

→ w^+ should go down and w^- should go up, so d^- response to CS+ should go down with training

So this actually goes against RPE coding by DANs.

More generally - in light of the overall lack of experimental evidence for RPE-coding by DANs in the fly mushroom body, what would it mean for the models in this manuscript to be falsified experimentally? I mean this in two senses: (1) what kind of evidence would falsify the models, and (2) is it possible to argue that any RPE coding circuit must obey certain features of the models presented here, so that falsifying these features would mean that DANs do not encode RPEs?

Fig 5: suggest adding larval data:

Saumweber et al Nat Comm 2018

König ... Gerber Biol Lett 2019

Admittedly this model is based on the adult mushroom body, but the larva seems to have equivalent architecture of punishment/reward DANs, approach/avoidance MBONs, feedback from MBONs to DANs (see <https://www.biorxiv.org/content/10.1101/649731v1>)

Prediction 2 - VSlambda model. If the predicted sustained DAN response is not observed, couldn't this just mean that lambda is very high? Or that the reward is not "sufficiently large"? I'm not sure this is really a experimentally testable prediction, in the sense that a negative result is not informative. Are there any other more robust predictions that would support VSlambda over MV?

Prediction 3 - can the authors run the VSlambda and MV models on the Felsenberg et al extinction result, to see if either model can reproduce the result (ie that blocking aversive DANs during re-exposure to a previously rewarded odor, this time without reward, prevents extinction). Where would these points go on Fig 5d?

Prediction 5 - Could MBONs of a single valence be modulated by DANs of both valence via cross-compartment interactions? - e.g. the inhibitory MVP2/g1-pedc "approach" MBON that is depressed by aversive learning and also inhibits M4/M6, thereby allowing aversive learning to potentiate M4/M6 odor responses to CS+?

Related to this: as the authors write in the Discussion, ref 10,14 shouldn't be cited in line 195 as examples of learning-induced potentiation (rather they are learning-induced depression of feedforward inhibition)

line 112 learning rate must be small to average over many odors. Yet animals learn after only a single trial. This discrepancy should be discussed.

line 165-66, 292: wk synapses from KCs to DANs are well-justified anatomically, but ref. 29 doesn't really show that KC-DAN synapses are required for learning. It shows that learning is impaired when nAChRs are knocked down in DANs, which could easily be because the R inputs to D are cholinergic. This alternative interpretation should be acknowledged

line 317 says the MV model correlates with experiments $R=0.63$ but the legend to Fig 5 says $R=0.44$ (a big difference!) - if the latter is true, it seems then that the MV model is not as good a fit to experiments as the VSlambda model.

Math points - possible typos (or I couldn't follow the derivation):

eq 2, 16 - where does the $1/2$ come from?

eq. 4 - derivation of VS plasticity rule

Possible typo: according to the derivation in the methods (eq 17-18) should eq 4 say $C+ = 1/2 \sum (r- + m+)^2$? (vs. what is written: $C+ = 1/2 \sum (r+ + m-)^2$?

derivation of eq 18

Is there a missing minus sign on the second step? i.e. $dCvs/dw+ = k(r- + w+*k)$, thus $P+ = -\eta*k*(r- + w+*k)$?

then the minus sign comes out in the 3rd step because $d- = r- + w+*k + wk*k$

Minor points:

line 219 typo: "combing"

Reviewer #3 (Remarks to the Author):

The authors develop a computational model of reward learning in the *Drosophila* mushroom body (MB) which learns through the minimization of reward-prediction errors (RPEs). They explore the shortcomings (i.e. learning limitations) of this model when it is constrained to be encapsulated in an architecture which is convergent with the currently agreed upon anatomy of the MB, and they then propose a new architecture which overcomes these limitations and offers new anatomical and physiological hypotheses for the MB.

General Impression

The authors have done a nice job of developing a method by which the MB could learn via RPEs as opposed to absolute reward and also explored well the limitations of this model. It finds that the furthest extension of the model (i.e. that which currently strays the furthest from known anatomy) best replicates certain physiological properties of the MB output neurons and dopamine neurons. They also validate the behavioral performance of this model and reduced model (which is in better agreement to known anatomy) and find that both make reasonably good statistical predictions of experimental behavioral data. Finally, the authors lay out 5 predictions between the two models which can be tested experimentally, offering a way to validate not just the importance of RPE signaling in the MB, but also to differentiate between the two models. I would recommend the paper for publication as is, with only one minor suggestion regarding Figure 5.

Critiques

1. As written the manuscript is not fully suitable for general neuroscience community. Lots of terminology is coming from machine learning literature, which is not bad by itself but will put limitations on who will/can read this paper. The authors should revise the text to include explanations of the model and results suitable for general reader.

2. Figures 5d and 5e show the fit of each model to experimentally obtained behavioral data. While the correlation coefficient is certainly important to assess the fit of the model predictions to the data, the slope of the regression line should be reported, as a slope near 1 would indicate that the models are not just making self-consistent predictions which covary tightly with the experimental data (i.e. a good qualitatively predictive model), but also can produce accurate quantitative predictions which can be directly compared to experiment.

Response to reviewers' comments

We are very grateful for the thoughtful and constructive comments from the three reviewers. We provide responses to each comment below, highlighted in bold font. We would also like to emphasise here the conceptual advance we believe our work makes. First, we identify how the activity of – and connectivity between – different mushroom body (MB) neuron populations can be attributed to different elements of reward prediction error (RPE) coding, and show for the first time how these elements may fit together in the MB. We do this following a normative approach, which allows us to describe precisely how RPE coding may take place in the MB, what possible limitations to RPE coding there may be, and to attach functional importance to different elements of the MB circuit that will enable strong predictions to be made for future experiments. Second, we provide predictions (including those described in the supplemental document provided), that we believe will shed light on the controversy surrounding the RPE hypothesis as applied to the MB, and suggest aspects of MB function that would need to be tested in order for the RPE hypothesis to finally be put to bed, or adopted.

Reviewer #1 (Remarks to the Author):

The authors describe a computational model intended to simulate neuronal circuits underlying associative learning in *Drosophila melanogaster*. It is well-established that the neuronal circuitry mediating associative learning is localized to the mushroom body of the *Drosophila* brain. This circuit is subject of intense research and serves as a general model system for analyzing principles of neuronal coding and plasticity. Therefore, the overall topic is certainly of interest not only for specialists working on *Drosophila*, but also for a more general scientific readership.

Here, it is proposed that computing prediction errors, a phenomenon well-known for the dopaminergic system of the mammalian brain, might also underlie associative learning in the mushroom body of the *Drosophila* brain. The idea that minimizing prediction errors through learning is not only present in mammalian brains but also in insects is not new and intriguing (see, for example, Hammer, 1997, *TiNS* 20, No. 6, 1997). This reviewer is not in the position to judge the mathematical details of the computational model itself. However, I am most comfortable with the details of the circuitry and the behavioral aspects. I like the overall approach taken by the authors and the concept of integrating positive reinforcement (reward) with negative reinforcement (punishment) and feedback loops. However, I would like to raise several points of concern the authors might want to consider.

First, experimental evidence for prediction error coding in *Drosophila* is lacking. A major argument for prediction error coding in mammals is the phenomenon of “blocking”. In this case, learning does not take place if the learned outcome is already predicted by a sensory cue. In insects, several researchers have searched for this phenomenon, but without any success. In honey bees, blocking has been reported, but later on this finding has not been confirmed. Therefore, from a behavioral point of view it is not clear whether any prediction error is actually computed and needed. The authors should provide at least arguments why learning through prediction errors must be inevitably present. As it stands, I do not see any reason to assume that prediction error coding in the insect brain is actually present.

The blocking phenomenon is regarded as a good indicator of RPE coding (Rescorla & Wagner, 1972), but its absence cannot unequivocally be taken as evidence against RPE coding. Indeed, the absence of blocking has also been reported numerous times in vertebrates, yet RPE coding remains one of the prominent

hypotheses for dopaminergic function in the vertebrate brain. Why is this? Rescorla & Wagner make the strong assumption that neural responses to the sensory cues that predict reinforcement are independent, which typically is the case when multimodal cues are used, but is not necessarily the case when unimodal cues are used. The differences between experiments that use unimodal and multimodal stimuli have been addressed extensively in [Soto, Gershman & Niv, (2014), Psych. Rev. 121, 526-558] and [Soto (2018), J. Exp. Psych.: General 147, 597-602] to explain those cases in which blocking was not observed in vertebrates, and we believe this provides a good explanation for the lack of strong evidence for blocking in *Drosophila* [Brembs & Heisenberg, (2001), J. Exp. Biol. 204, 2849-2859] and bees [Gerber & Ullrich (1999), J. Exp. Biol. 202, 1839-1854; Guerrieri et al. (2005), Learn. and Mem. 12, 86-95]. Even among earlier studies in which blocking was reported in bees, the blocking effect was not complete, but partial [Smith (1997), Behav. Neurosci. 111, 57-69]. In cockroaches, complete blocking has been observed using multimodal stimuli (Terao et al. (2015), Scientific Reports 5, 8929). Our model is able to capture complete blocking, partial blocking, or the absence of blocking, depending on the extent to which single stimulus representations in Kenyon cells are altered during the compound stimulus training phase (details in the attached document). Although we were planning to present these results in a follow up manuscript, we believe the reviewers' concerns about the blocking phenomenon warrants their inclusion in this manuscript, and that they will also be of substantial interest to the insect community.

Second, the authors postulate a feedback from MBONs back to DANs. Currently, there is enough evidence that this feedback is actually present (e.g., Ichinose et al., Elife 2015, 4:e10719). One does not have to postulate it. It has already been postulated by Riemensperger et al. (2005), cited by the authors. However, recent connectomics studies have determined massive connections between KCs and DANs (reciprocal synapses) (see also Cervantes-Sandoval et al., Elife 2017, 6: e23789) and, moreover, massive direct feedforward connections between DANs and MBONs (the EM-connectomics studies cited). These connections do not appear in the authors' models. I recommend that the authors base their models on the data from the most recent connectomics studies.

We believe the reviewer may have confused the known connectivity we are drawing on to build our model with the connectivity we are postulating. In paragraph 3 of the introduction (page 3), we cite 4 papers as evidence for feedback from MBONs to DANs, which we use to build our first instantiation of the model. What we do postulate, however, is that the class of MBON and the class of DAN determines whether the connection between them is excitatory or inhibitory. We state this for the first time in the first new paragraph on page 8. We will clarify this in the manuscript by better emphasising what we are postulating, and what we are using as fact.

Regarding reciprocal KC-DAN connections, these do appear in our model in the form of excitatory KC->DAN connections, and modulatory DAN->KC connections (as dopamine-dependent synaptic plasticity between KCs and MBONs is thought to occur presynaptically by modulating the release probability).

Regarding DAN->MBON connections, please refer to our response to comment 4 below.

Minor remarks:

1. Abstract: the authors write "...and use simulations to verify that MBONs learn accurate predictions." This is an overstatement. A simulation can only postulate this. Verification must be done experimentally.

We will replace this with “...use simulations to verify that MBONs in the model learn accurate predictions”.

2. The authors often write about reward. In *Drosophila*, experiments are often done using punishment as reinforcement. Perhaps, the authors should verbalize very early in the introduction that in their models punishment reflects negative reward.

We state in ‘Methods: Experimental Paradigm’ that we refer to punishments as negative rewards, so that we can continue with the familiar expression of “reward prediction error”. We will also add this point to the Results section for clarity.

3. Introduction, line 63: The authors write “To the best of our knowledge, these latter two features have yet to be incorporated in computational models of the MB).” Okay, but why do these two features HAVE to be integrated if there is no evidence that reward prediction error coding exists in insects?

We incorporate MBON->DAN feedback to postulate a role for these connections, and to align our model more closely with the known anatomy. Because some form of feedback is required for RPEs to be computed, such connections increase the plausibility of the RPE hypothesis as applied to DANs in the mushroom body. As outlined in the Introduction of our manuscript, we believe there is additional, indirect evidence for RPE coding in the mushroom body. The rest of the paper aims to demonstrate what RPE coding in the mushroom body might look like in order to motivate new experiments that may better determine a conclusion to this hypothesis. Please also refer to our response above regarding the evidence for/against blocking in insects.

4. Introduction, line 72: Feedback from MBONs onto DANs is nowadays well-described. Given that the MB connectome of the larval MB and parts of the adult MB is available one can assume that all possible connections are known. When the first connectomics data were published the community was astonished about the feedforward connection from DANs onto MBONs. Do the authors see any possibility to make sense of that?

Currently, we cannot propose a role for these connections in RPE coding using our current model. The authors who first reported this connection (Takemura et al. (2017), eLife 6:e26975) proposed that recurrent excitation between DANs and MBONs could serve two roles: 1) to sustain MBON activity and, therefore, sustain the motivational bias induced by the MBON during behaviour; 2) in order for learned rewards to be compared with current rewards, such that behaviour is driven by predictions for the rate of change in reward. However, in our current model, we use a simplified paradigm and do not address learning or behavioural dynamics within a trial, but only between trials. As such, we do not yet have a hypothesis for how sustained MBON activity, or reward comparisons over time scales of just seconds (the rise/fall time of DAN->MBON synaptic potentials) would be relevant to our paradigm of learning over time scales of minutes to hours.

5. Results part: This part is nicely explained. In fact, I really like how the authors describe their models and the rationale behind it. And the figures provide very good illustrations.

Greatly appreciated.

6. Discussion, line 369: the authors write “to the best of our knowledge, only one study has measured DAN responses throughout learning in *Drosophila*”. This is factually wrong. Please have a closer look at Mao and Davis, *Front Neural Circuits*. 2009, 3:5, and Riemensperger et al., (2005) (cited). In the latter publication, it was explicitly searched

for prediction error coding, but with a negative outcome. I think these data should not be ignored.

We will update this statement to make clear that we are referring to measurements of DAN activity after each trial throughout training. Mao & Davis, and Riemensperger et al., only show DAN responses before and after training, as opposed to throughout training. We will refer to these findings in the discussion, and describe in more detail how they fit alongside our model in the Supplementary Information, including the figure we have presented in the attached document.

Overall, I recommend that the authors incorporate these ideas in an updated version of the manuscript. The comments are not intended to unduly criticize the study but to help to improve it. However, I do not see any reason to believe that prediction errors exist in insects, and there are data that actually speak in favor that they do not.

In vertebrates, RPE coding by dopamine is the dominant theory for reinforcement learning. As the reviewer correctly points out, several studies have cast doubt on RPE coding in insects by addressing the blocking phenomenon and by measuring DAN activity. Unfortunately, these studies overlook some of the complexities of reinforcement learning algorithms (for example, the discount factor in temporal difference learning) and their implementation in realistic neural networks (for example, the neural representation of compound stimuli in blocking experiments, and the relevance of their being unimodal/multimodal - note also that complete blocking has been observed in insects when multimodal stimuli were used: see Terao et al. (2015), as cited above). Using our model, we are able to explain away these doubts, explain several additional characteristics of mushroom body function that have been observed in experiments, attach computational functions to known connections, and to predict additional connections that ought to support RPE coding (such connections have since been reported, though are not yet published: [Eschbach et al. (2019), bioRxiv, DOI: <http://dx.doi.org/10.1101/649731>]). We therefore believe that our model provides the first formal presentation of the RPE hypothesis applied to the mushroom body that, we hope, will encourage new experiments, the outcomes of which will either help to properly put this hypothesis to bed, or to adopt it.

Reviewer #2 (Remarks to the Author):

Bennett et al develop several potential computational models of the Drosophila mushroom body positing that it computes reward prediction error. While there has been substantial interest in the field in the general idea of the mushroom body as computing prediction error, no one has fleshed out the idea in detail, certainly not to the point of developing computational models that generate specific experimental predictions. By doing so, this manuscript makes a substantial contribution to the field. Therefore, I support publication in principle - but there are some points where the models should be clarified, better motivated, or indeed perhaps altered.

Conceptual points:

My biggest confusion is that I couldn't follow the reasoning leading up to the development of the MV model. It seems that the model in Fig 3a solves the bounded learning problem in the VSlambda model and the only experimental difficulty is that real DANs are excited, not just inhibited, by rewards/punishments. Why not just solve this by adding only both r+ and r- as inputs to DANs, rather than adding all the extra things in Fig 3c which haven't been observed experimentally? so then $d^- = r^- - r^+ + w^+ * k + w^- * k$

I assume there is some reason why this doesn't work, but this is never explained (or if it was, I didn't understand it).

In this case, D+ exhibits absolute reward responses to r+, and RPE responses to r-. Similarly, D- exhibits absolute responses to r-, and RPE responses to r+. Such a model is a combination of the first version of the VS model we present, which we show cannot learn, and the first VSu model (with unbounded learning, Fig. 3a). Now, when reinforcements are positive, D+ does not contribute to learning, but D- does, and when reinforcements are negative, D- does not contribute to learning, but D+ does. Thus, although the excitatory reward inputs do modulate DAN activity, they serve no role in learning. In effect, it behaves exactly the same as the model in Fig. 3a.

Similarly, it seems to me that the original RPE cost function and plasticity rule:

$$P(\text{RPE})_+ = \eta * k * (r_+ - r_- - (w_+ * k - w_- * k))$$

could be satisfied without requiring that D- has to potentiate M- as well as depressing M+ (the problematic Prediction 5). Couldn't you only suppose that D- is excited by R-, inhibited by R+, excited by M+ and inhibited by M-? that is: $d_- = r_- - r_+ + w_+ * k - w_- * k + w_k * k$

If this is impossible or fails some other criterion, I would like to see why.

In general, the MV model is a lot more complicated than the VSlambda model so I would like to understand what benefit each new complication brings.

This is a helpful observation. Such a model will indeed implement P(RPE)+/- exactly, and provides a more simple model that does not require D+/D- to potentiate M+/M-. Notably, this has no quantitative effect on the behaviour of the MV model (except for a factor of 2 in amplitude of synaptic weight updates, which is a minor difference that can easily be accommodated in the learning rate). We could therefore remove Prediction 5.

Prediction 1

line 381 - Dylla et al observe increased response to CS+ in DANs. But this is not consistent with either the VSlambda or the MV model.

VSlambda: $d_- = r_- + w_+ * k + w_k * k$

—> w_+ should go down with training, so d_- response to CS+ should also go down with training

MV model: $d_- = r_- - r_+ + w_+ * k - w_- * k + w_k * k$

—> w_+ should go down and w_- should go up, so d_- response to CS+ should go down with training

So this actually goes against RPE coding by DANs.

Dylla et al observe a decrease in responses to the US, and an increase to the CS+. For two reasons, it is only the US response to which our model, in its current form, can be compared. 1) We are not modelling neuronal activity throughout the course of each trial, and therefore do not distinguish between CS+ and US responses in isolation. 2) The plasticity rule implements the Rescorla-Wagner (RW) learning rule, which will not produce an increase in the CS+ response. Temporal difference learning, of which the Rescorla-Wagner learning rule is a simplified version, does yield increases in the CS+ responses, as well as the decreases in the US response that we capture in the present model. In 'Discussion: Prediction 1', we state that the Rescorla-Wagner rule, which we use in this work, can be extended to produce the temporal difference learning rule. We will also clarify that the RW rule cannot, however, produce the increase in the CS+ response.

More generally - in light of the overall lack of experimental evidence for RPE-coding by DANs in the fly mushroom body, what would it mean for the models in this manuscript

to be falsified experimentally? I mean this in two senses: (1) what kind of evidence would falsify the models, and (2) is it possible to argue that any RPE coding circuit must obey certain features of the models presented here, so that falsifying these features would mean that DANs do not encode RPEs?

In response to point 1) The strongest refutation of the model would be to demonstrate that there is no pair of single PPL1/PAM DANs for which the difference between their firing rates decays over time during repeated presentations of a CS with an initially unexpected US. However, this is a lot to ask of an experiment given current techniques. A more reasonable test would be to show that no single DAN, by itself, exhibits responses to an initially unexpected US that asymptote towards a baseline firing rate. However, such signals may be obscured by inputs that are not related to reinforcement learning but are common to both appetitive and aversive DANs. Furthermore, given the difficulty of recording from single neurons in the mushroom body, another test would be to show that the difference in Ca^{2+} signals between subpopulations of appetitive-coding and aversive-coding DANs does not decay after repeated exposures to an initially unexpected US. However, this risks missing the RPE signal as a result of the heterogeneous functional properties of DANs within these two classes. Furthermore, it is important that US responses are always cued by the CS. Riemensperger et al. (2005), for example, compare the US response in the final trial of training to the US response in the absence of a CS before training began. This was not the correct comparison to make when testing for RPEs, as the US-predicting cues were not consistent before and after training.

In response to point 2) Unfortunately, as there are so many ways by which RPE coding could be implemented by an unconstrained neural network model, it would be very difficult to identify a feature that would have to be common to all neural circuits that compute RPEs. We wish to emphasise that our contribution here is to ask what RPE coding might look like in the mushroom body, and to what extent it can explain the experimental data to date. A key ingredient would be for DANs to receive a copy of the US prediction. We postulate that this comes from the MBONs, but it is of course possible that US prediction may come from elsewhere in the brain. Thus, to be clear, our work concerns the idea that DANs encode RPEs in conjunction with the idea that MBONs encode reward predictions. If changes in MBON activity (more specifically, changes in the difference between approach and avoidance MBON activity) were not correlated with changes in DAN activity, this would be a good indicator that MBONs are not encoding the reward predictions necessary for DANs to compute RPEs, if DANs are indeed computing RPEs at all. Of course, there is now ample experimental evidence to show that changes in DAN activity are indeed strongly correlated with changes in MBON activity.

Fig 5: suggest adding larval data:
Saumweber et al Nat Comm 2018

We will cite this paper when addressing the connectomics data.

König ... Gerber Biol Lett 2019

Admittedly this model is based on the adult mushroom body, but the larva seems to have equivalent architecture of punishment/reward DANs, approach/avoidance MBONs, feedback from MBONs to DANs (see <https://www.biorxiv.org/content/10.1101/649731v1>)

We will add a citation to this work in the Discussion. The larval data (Eschbach et al. (2019), following the bioRxiv link) provides evidence for cross-compartmental connections from MBONs to DANs, as predicted in our model. The model presented in the Eschbach et al. paper is also noteworthy, though takes a different approach to our work, in that they tune the functional properties given a strict

architecture, rather than proposing architecture that achieves a particular function, as we have done.

Prediction 2 - VSlambda model. If the predicted sustained DAN response is not observed, couldn't this just mean that lambda is very high? Or that the reward is not "sufficiently large"? I'm not sure this is really a experimentally testable prediction, in the sense that a negative result is not informative. Are there any other more robust predictions that would support VSlambda over MV?

The converse of Prediction 4 would support the VSlambda model. I.e. the prediction would be that learning is mediated by changes in the firing rate of either approach MBONs or avoidance MBONs, but not both at the same time. This could be incorporated into Prediction 4 for clarity.

Prediction 3 - can the authors run the VSlambda and MV models on the Felsenberg et al extinction result, to see if either model can reproduce the result (ie that blocking aversive DANs during re-exposure to a previously rewarded odor, this time without reward, prevents extinction). Where would these points go on Fig 5d?

We can incorporate these data in the Supplementary Information. Briefly, our model supports the results from the Felsenberg et al. (2017, 2018) papers. For comparison with the 2017 paper:

- 1) Blocking D- during CS+ reactivation after appetitive conditioning abolishes extinction (because blocking D- acts as an appetitive learning signal) - we would expect the data point to fall in the centre of the graph.**
- 2) Blocking D- during CS- reactivation results in extinction of approach to the CS+ because the CS- acquires a positive valence - we expect the data point to fall in the lower left corner of the graph.**
- 3) Blocking D+ during CS+ reactivation results in a preference for the CS-, i.e. going beyond extinction (because blocking D+ acts as an aversive learning signal) - we expect the data point to fall even further toward the lower left corner of the graph.**

Similar results are obtained for aversive conditioning, as in the 2018 paper.

Prediction 5 - Could MBONs of a single valence be modulated by DANs of both valence via cross-compartment interactions? - e.g. the inhibitory MVP2/g1-pedc "approach" MBON that is depressed by aversive learning and also inhibits M4/M6, thereby allowing aversive learning to potentiate M4/M6 odor responses to CS+?

We can add discussion about such an interaction to the manuscript. This indeed could be incorporated into our model, and could surely be made to work as has been shown in experiments. It would be interesting to see whether MVP2 has a similar effect on plasticity at V2 MBONs. However, to date, experiments seem to suggest there is no such symmetry in the MVP2 function.

Related to this: as the authors write in the Discussion, ref 10,14 shouldn't be cited in line 195 as examples of learning-induced potentiation (rather they are learning-induced depression of feedforward inhibition)

We will update this line to clarify this point.

line 112 learning rate must be small to average over many odors. Yet animals learn after only a single trial. This discrepancy should be discussed.

This does at first seem at odds with requiring a sufficiently small learning rate. However, changes in behaviour may still be observed, and recognised as learning having taken place, even if learning is not complete. In our model, "complete" would mean that the RPE has decayed to zero. We will clarify this point in the manuscript.

line 165-66, 292: wk synapses from KCs to DANs are well-justified anatomically, but ref. 29 doesn't really show that KC-DAN synapses are required for learning. It shows that learning is impaired when nAChRs are knocked down in DANs, which could easily be because the R inputs to D are cholinergic. This alternative interpretation should be acknowledged

We have updated the manuscript to describe this alternative explanation for the impaired learning observed in ref. 29.

line 317 says the MV model correlates with experiments $R=0.63$ but the legend to Fig 5 says $R=0.44$ (a big difference!) - if the latter is true, it seems then that the MV model is not as good a fit to experiments as the VSlambda model.

This was a typo carried over from a preliminary run of our analyses, and has been corrected to show the true R value of 0.63.

Math points - possible typos (or I couldn't follow the derivation):

eq 2, 16 - where does the $1/2$ come from?

This is a typo and has been corrected (the $1/2$ does apply, however, for the MV model, as $d_+ - d_- = 2(r_+ - r_- - (m_+ - m_-))$).

eq. 4 - derivation of VS plasticity rule

Possible typo: according to the derivation in the methods (eq 17-18) should eq 4 say $C_+ = 1/2 \sum (r_- + m_+)^2$? (vs. what is written: $C_+ = 1/2 \sum (r_+ + m_-)^2$?)

Yes indeed. Corrected.

derivation of eq 18

Is there a missing minus sign on the second step? i.e. $dC_v/dw_+ = k(r_- + w_+*k)$, thus $P_+ = -\eta*k*(r_- + w_+*k)$?

then the minus sign comes out in the 3rd step because $d_- = r_- + w_+*k + wk*k$

Thanks, typo corrected.

Minor points:

line 219 typo: "combing"

Corrected.

Reviewer #3 (Remarks to the Author):

The authors develop a computational model of reward learning in the Drosophila mushroom body (MB) which learns through the minimization of reward-prediction errors (RPEs). They explore the shortcomings (i.e. learning limitations) of this model when it is constrained to be encapsulated in an architecture which is convergent with the currently agreed upon anatomy of the MB, and they then propose a new architecture which overcomes these limitations and offers new anatomical and physiological hypotheses for the MB.

General Impression

The authors have done a nice job of developing a method by which the MB could learn via RPEs as opposed to absolute reward and also explored well the limitations of this model. It finds that the furthest extension of the model (i.e. that which currently strays the furthest from known anatomy) best replicates certain physiological properties of the MB output neurons and dopamine neurons. They also validate the behavioral performance of this model and reduced model (which is in better agreement to known anatomy) and find that both make reasonably good statistical predictions of

experimental behavioral data. Finally, the authors lay out 5 predictions between the two models which can be tested experimentally, offering a way to validate not just the importance of RPE signaling in the MB, but also to differentiate between the two models. I would recommend the paper for publication as is, with only one minor suggestion regarding Figure 5.

Critiques

1. As written the manuscript is not fully suitable for general neuroscience community. Lots of terminology is coming from machine learning literature, which is not bad by itself but will put limitations on who will/can read this paper. The authors should revise the text to include explanations of the model and results suitable for general reader.

We will endeavour to provide a more general description at least once for technical terms, and will consider replacing technical jargon where possible.

2. Figures 5d and 5e show the fit of each model to experimentally obtained behavioral data. While the correlation coefficient is certainly important to assess the fit of the model predictions to the data, the slope of the regression line should be reported, as a slope near 1 would indicate that the models are not just making self-consistent predictions which covary tightly with the experimental data (i.e. a good qualitatively predictive model), but also can produce accurate quantitative predictions which can be directly compared to experiment.

We can provide the slope of the linear fit. However, unfortunately, there is currently such variability between experimental data sets, even when the same or a similar experimental protocol is used, that the accuracy with which the model can make quantitative predictions for individual experiments will be limited.

What is reassuring, however, is that both the VSlambda and the MV models yield, for the most part, very consistent predictions to each other (there is the odd exception, as noted in the results, and we analyse such cases in the Supplementary Information as a means of distinguishing between the behaviours of both models). Plotting the data from the VSlambda model against that from the MV model yields a regression with a slope very near to 1.

Reviewer #1 (Remarks to the Author):

The authors have spent considerable effort to address the points of concern raised by the reviewers. In particular, they have included some arguments why a lack of behavioral blocking in insects does not argue against reinforcement prediction error coding in DANs. Overall, the manuscript has improved. But there is one point that I disagree with and that needs to be clarified: a lack of evidence for reinforcement prediction error coding still exists from physiological experiments. This should be spelled out.

The authors have included a strange supplemental discussion section in which they argue, in a rather dismissive way, that an analysis reported by Riemensperger et al. (2005) was incorrect or inappropriate. It must be noted that Riemensperger et al. provided the first physiological study at all indicating that dopaminergic neurons carry reinforcing properties in insects. In this light it is unfair and inappropriate to state that:

"Appropriate comparison of DAN responses before and after conditioning

To test for changes in DAN responses to the US, Riemensperger et al. compared the US response in the final trial of conditioning, during which the US presentation follows the CS+ stimulus, to the US response in isolation before conditioning begins. Thus, the stimuli preceding the US differed between the two US responses (either being absent or present), yet this is not the correct comparison to test the predictions of TD learning."

This paragraph is actually what one could call "handwaving". Even if the CS+ preceding the US would influence the calcium signal evoked by the US, there is no evidence for prediction error coding in this particular experiment. Prediction error coding would require that the response to the US decreases, even if a CS+ precedes. In addition, the response to the predicted CS+ should increase, which is not the case. One can of course argue that the population of dopaminergic neurons is now known to be more heterogeneous than previously known, or that calcium imaging has a lower spatial and temporal resolution than spike rate measurements would provide. But degrading the only physiological studies available in a supplemental discussion just to fit in their own theoretical argument is scientifically questionable. Please correct that and discuss all scientific evidence in a positive, fair and constructive way!

Reviewer #2 (Remarks to the Author):

The authors have largely addressed my concerns. A particularly important improvement is how they've addressed Reviewer 1's concern about blocking, which allows this manuscript to make an additional important contribution to our theoretical understanding of how to link prediction errors

to behavioral experiments.

I just have one remaining issue. I did not understand the authors' reply to my point about Prediction 1 - that Dylla et al 2017 observed increased responses to CS+ in DANs after training while the authors' VSlambda and MV models predict decreased responses to CS+.

I understand the authors' argument that the Rescorla-Wagner rule does not speak to neural responses to the CS. And I understand that the authors' simulations always present the CS and US together, so given their equations

VSlambda: $d^- = r^- + w^+ * k + w_k * k$

MV model: $d^- = r^- - r^+ + w^+ * k - w^- * k + w_k * k$

it makes sense that, as the authors say, during training, d-'s response to the CS+US together initially goes up (because r- is suddenly higher) but then goes down as training progresses (because w+ goes down).

But I don't understand the authors saying that DAN responses to the CS+ alone will not change in their model. Maybe changing DAN responses to CS+ are not a normal prediction of RW rules, but a surface reading of the equations indicates that if r- and r+ are 0 (no US presented, i.e. only CS is presented), then d- activity is determined by w+, w-, and wk. Therefore if w+ decreases (as it would after learning), then the DAN's response to CS+ alone (without US) would be smaller after training than before. And this is exactly what Dylla et al do not observe when they present CS+ alone, before and after training.

I would be happy for the authors to either explain how I've misunderstood, or speculate on reasons why the experiments don't match the theory, as they did for Dylla et al.'s findings that the DAN response to the US is the same for both paired and unpaired conditions (lines 466-473).

Reviewer #3 (Remarks to the Author):

The authors present a well-done theoretical experiment to investigate how RPE might be represented and used to guide learning. These models are also compared favorably to a large set of experimental results. Finally, the authors present multiple testable predictions that could help validate their mechanism of RPE-based learning. I found this work to be interesting and valuable to a broad neuroscience community. Nevertheless, there are issues that need to be addressed, as summarized in my comments below.

GENERAL COMMENTS

VSλ model with adaptation: One of the main findings of this work is that the simplest model VSλ is limited by upper bounds in the reward tolerance, too much reward (punishment) results in sustained RPE. However, a simple solution (which the authors provide) is adaptation. This is only discussed briefly and in the end of the discussion. It seems likely that DANs could have mechanisms to adapt baseline firing rates over long periods (such mechanisms are pervasive in other types of neurons). The concept is raised but not explored. How would the VSλ-adapt model fair in the experiments shown in Figure 5?

Lack of comparison with non-RPE model: As the authors have mentioned, there is a large community

that disputes the possibility that DANs in insects actually represent RPEs, instead DANs may represent absolute reward values. However, there is no comparison or discussion of predictability or benefits of an RPE-based vs a reward-level based DAN mechanism.

SPECIFIC COMMENTS

In abstract and throughout RPs, RPEs, and RPTs are so similar, does the paper need all these abbreviations? It can be confusing when one reads RPs and expect RPEs

Abstract: “[we] postulate as yet unobserved connectivity that overcomes limitations in the experimentally constrained model.” – This is vague and it not clear if what you postulate could be useful.

Line 34 & 39: blocking phenomenon has not been introduced. What is blocked? Why?

Line 42: “Consistent with...” complex sentence and it is unclear what is supposed to be consistent with it’s role: organization or innervation. Rephrase.

Line 47 & 51: protocerebral anterior medial (PAM) cluster; parentheses around abbreviation not the other way consistent with MBONs, DANs, etc.

Line 51: Cite figure 1c.

Figure 1a Caption: “The red region highlights...” should say green.

Around eqn(2) – say that k is the firing rate of the KC population

Line 157: shouldn’t this be $r+m>0$ not $r>0$ or $m>0$?

Figure 2. I am sort of surprised the authors don’t plot w^- and w^+ . As this paper concerns the mechanisms of learning, it would be interesting to see the changes in synaptic strength.

Figure 2b-d Caption: say explicitly in the caption that $\gamma = 1$.

Line 192: one more sentence on the consequences for learning. Or does it matter? Since in these two cases (when reinforcement exceeds upperbound), one of M^{\pm} is zero and the correct action should certainly be taken.

Line 212: It’s probably worth explaining this in a little more detail as it motivates Figures 3b-d. I found it difficult to understand why the model in 3a couldn’t replicate appetitive vs aversive DANs. Really, the authors mean DANs should respond to the appearance (not removal) of reward. The inhibitory model in 3a only has DANs responding to the removal of a reward. Is this correct?

Line 254: Explain why there are two solutions... essentially one comes from cost in (1) vs (4)?

Line 269: “when w^{\pm} changes reach zero”, you don’t show w^{\pm} . If the authors discuss an explicit value of w you need to provide plots of w^{\pm} explicitly.

Figure 3e: The plots show an increase of the baseline firing rates (zero RPE firing rates) of the MBONs after each cycle of reward changes (compare trials [0-20], [80-100], and [160-180]). What is the mechanism of this? Are they unstable, i.e. will a continuation of the reward changes lead to a growth of the MBON baseline activity?

Line 275: the authors might mention: blocking KC->DAN synaptic transmission (by setting $\gamma=0$). Not at all clear which model is plotted (VSL vs MV) in Figure 4.

Lines 299-303: Very precise statement with no data shown. Also, it is unclear why (even if it is in the Supp Fig 6 which I don’t have) this would occur. More discussion (and figures) are necessary.

Figure 5: $x=0$ and $y=0$ lines would be useful.

Figure 5: Slopes are small indicating that the model was more sensitive to perturbations than animals. This needs discussion.

Lines 380 – 385: Again, “blocking” is used without a definition in the abstract and introduction. The precise meaning of this term might not be obvious for the wide audience of Nature comm (it wasn’t for me). These lines should be at least given in the introduction. I wouldn’t mention blocking in the abstract. Also, I would try to rephrase a little because X comes first (conditionally) then XY then Y.

You might at least add “the subsequent association between Y and R...” to drive this sequential process home.

Figure 6e. The x-axis labels are unclear. You might want to say $p_X = 0.8$; $p_Y = 0.2$ with explanations in caption? Or independent vs non-independent? Instead of two very similar honeycombs.

Prediction 2: Line 486: “If w_k^T cannot be determined, then any...”. What does “cannot be determined” mean? The authors seem to imply that it means that KCs have no effect on DANs (i.e. w_k^T doesn’t affect the equation for plasticity), which seems like an extreme difference from the VSL model. What is really discussed here?

Prediction 2: It seems that adaptation of baseline firing rate to long-term reward signal could account for much of this limitation. Since this may solve a complicated connectivity issue (i.e. lend support to VSL over MV models), it seems this is worth exploring.

Line 565: It would be relevant to cite the Cassenear and Laurent papers from 2012 and 2007 in locusts.

Prediction 3: The valence of DANs is defined by its response to RPEs not reinforcement – This is, perhaps, the biggest conclusion of the work on the MV model. I would add text to highlight this in the results and potentially abstract. Perhaps also allude to other animal models where neurogenic amines are used for RPEs.

Prediction 5 and prediction 3 are connected: 5 is a consequence of 3.

Line 688: Shouldn’t m_q have a w_m coefficient?

Line 704: More discussion is required (in the main text as well) as to why two solutions are presented. It is entirely unclear how (19) relates to (20) as it seems you’ve set $w_K = [1, 1, \dots, 1]$. Does “or” for eqn (19) mean there’s an alternate derivation that is not shown or discussed? Is it supposed to be the same thing? (I highly doubt this). In any case, more discussion is necessary in the methods and only one (the one you actually use) should be presented in the main text.

We wish to thank the reviewers for their critical and constructive feedback. Please find our responses below in **bold text**.

Reviewer #1 (Remarks to the Author):

The authors have spent considerable effort to address the points of concern raised by the reviewers. In particular, they have included some arguments why a lack of behavioral blocking in insects does not argue against reinforcement prediction error coding in DANs. Overall, the manuscript has improved. But there is one point that I disagree with and that needs to be clarified: a lack of evidence for reinforcement prediction error coding still exists from physiological experiments. This should be spelled out.

The authors have included a strange supplemental discussion section in which they argue, in a rather dismissive way, that an analysis reported by Riemensperger et al. (2005) was incorrect or inappropriate. It must be noted that Riemensperger et al. provided the first physiological study at all indicating that dopaminergic neurons carry reinforcing properties in insects. In this light it is unfair and inappropriate to state that:

"Appropriate comparison of DAN responses before and after conditioning

To test for changes in DAN responses to the US, Riemensperger et al. compared the US response in the final trial of conditioning, during which the US presentation follows the CS+ stimulus, to the US response in isolation before conditioning begins. Thus, the stimuli preceding the US differed between the two US responses (either being absent or present), yet this is not the correct comparison to test the predictions of TD learning."

This paragraph is actually what one could call "handwaving". Even if the CS+ preceding the US would influence the calcium signal evoked by the US, there is no evidence for prediction error coding in this particular experiment. Prediction error coding would require that the response to the US decreases, even if a CS+ precedes. In addition, the response to the predicted CS+ should increase, which is not the case. One can of course argue that the population of dopaminergic neurons is now known to be more heterogeneous than previously known, or that calcium imaging has a lower spatial and temporal resolution than spike rate measurements would provide. But degrading the only physiological studies available in a supplemental discussion just to fit in their own theoretical argument is scientifically questionable. Please correct that and discuss all scientific evidence in a positive, fair and constructive way!

We thank the reviewer for pointing out the clarifications that are needed in our argument. It was not our intention at all to discredit Riemensperger et al., but rather to describe the questions that still remain. We have attempted to rewrite this discussion so as to properly acknowledge what was achieved by the seminal work of Riemensperger et al. (predictive signals in the form of prolonged DAN responses to the CS+), as well as acknowledging explicitly that their work found no evidence for the signatures of RPE coding as predicted by TD learning. We then go on to discuss a possible reason for why the CS+ response does not increase (relating to the discount factor), as well as our description of the experimental difficulties in determining prediction error responses due to the heterogeneity of DAN responses. We have also added how the temporal and spatial precision of calcium imaging compounds these experimental difficulties. In addition, we incorporated a paragraph to address comments from Reviewer 2, which also relate to the CS+ response in DANs.

Reviewer #2 (Remarks to the Author):

The authors have largely addressed my concerns. A particularly important improvement is how they've addressed Reviewer 1's concern about blocking, which allows this manuscript to make an additional important contribution to our theoretical understanding of how to link prediction errors to behavioral experiments.

I just have one remaining issue. I did not understand the authors' reply to my point about Prediction 1 - that Dylla et al 2017 observed increased responses to CS+ in DANs after training while the authors' VSlambda and MV models predict decreased responses to CS+.

I understand the authors' argument that the Rescorla-Wagner rule does not speak to neural responses to the CS. And I understand that the authors' simulations always present the CS and US together, so given their equations

$$\text{VSlambda: } d^- = r^- + w^+ * k + w_k * k$$

$$\text{MV model: } d^- = r^- - r^+ + w^+ * k - w^- * k + w_k * k$$

it makes sense that, as the authors say, during training, d^- 's response to the CS+US together initially goes up (because r^- is suddenly higher) but then goes down as training progresses (because w^+ goes down).

But I don't understand the authors saying that DAN responses to the CS+ alone will not change in their model. Maybe changing DAN responses to CS+ are not a normal prediction of RW rules, but a surface reading of the equations indicates that if r^- and r^+ are 0 (no US presented, i.e. only CS is presented), then d^- activity is determined by w^+ , w^- , and w_k . Therefore if w^+ decreases (as it would after learning), then the DAN's response to CS+ alone (without US) would be smaller after training than before. And this is exactly what Dylla et al do not observe when they present CS+ alone, before and after training.

I would be happy for the authors to either explain how I've misunderstood, or speculate on reasons why the experiments don't match the theory, as they did for Dylla et al.'s findings that the DAN response to the US is the same for both paired and unpaired conditions (lines 466-473).

We apologise for our misunderstanding about the Reviewer's original question regarding Prediction 1, and thank them for clarifying. The Reviewer quite rightly points out that, in our models, DAN responses to the CS+ alone would decrease. We now discuss this question, and explain how it fits with the experimental data and theory in the Supplemental Text, line 122. Briefly, we think the confusion here can be resolved by viewing each response in our model as a combination of two components, which are predicted by TD learning theory and are measured in Dylla et al. (2017). The first component is the CS+ response, and the second component is the US (whether present or omitted) response. In Dylla et al., the CS+ response increases and the US response decreases throughout conditioning (though the latter is inconclusive, which we had discussed in lines 466-473, now on lines 477-487). Because RW learning does not change the CS+ component, any change in the model's DAN response is simply a result of change in the US component. In our model, omitting a punishing US causes an inhibitory response in D-, just as observed by Schultz et al. (1997) when reward was

omitted. Thus, when $r^- = 0$ and $r^+ = 0$ after a period of conditioning, this should be thought of as a response to an omitted US, rather than a response to the CS+. An alternative way of thinking about the case when $r^- = r^+ = 0$ after a period of aversive conditioning, is that the punishment signal has simply been reduced, rather than omitted, resulting in a positive RPE, signalled by a decrement in d^- (and increment in d^+ in the MV model). Although this inhibitory response to the omitted US is not observed in Dylla et al., the reason may be that the DAN firing rate has already decreased to near zero. Any further reduction in firing rate due to an inhibitory response would not be detected with calcium imaging.

Reviewer #3 (Remarks to the Author):

The authors present a well-done theoretical experiment to investigate how RPE might be represented and used to guide learning. These models are also compared favorably to a large set of experimental results. Finally, the authors present multiple testable predictions that could help validate their mechanism of RPE-based learning. I found this work to be interesting and valuable to a broad neuroscience community. Nevertheless, there are issues that need to be addressed, as summarized in my comments below.

GENERAL COMMENTS

VS λ model with adaptation: One of the main findings of this work is that the simplest model VS λ is limited by upper bounds in the reward tolerance, too much reward (punishment) results in sustained RPE. However, a simple solution (which the authors provide) is adaptation. This is only discussed briefly and in the end of the discussion. It seems likely that DANs could have mechanisms to adapt baseline firing rates over long periods (such mechanisms are pervasive in other types of neurons). The concept is raised but not explored. How would the VS λ -adapt model fair in the experiments shown in Figure 5?

The question of adaptation in DANs is certainly interesting, and one we are pursuing. Despite the advantages that adaptation may afford, the MV model nevertheless has the advantage of not requiring adaptation in order to learn, as well as being able to explain additional experimental observations (e.g. that both approach and avoidance MBON firing rates change after learning, or that inhibition of aversive DANs acts as reward – both discussed in the manuscript). For the purposes of Fig. 5, for the VS lambda model, we set lambda to be large, such that the reinforcements provided in our simulations (+1 for appetitive training, -1 for aversive training) did not exceed the upper bound. We now state this on line 820. As such, the limitations of the VS lambda model do not impose themselves on the results presented in Fig. 5. Furthermore, addressing considerations such as the mechanism and the time scale of adaptation would, in our opinion, go beyond the scope of this manuscript, the focus of which is about the feasibility of RPE coding in the mushroom body, and the explanatory power of the VS and MV models (the MV model explains a larger number of physiological results, at the expense of additional connectivity).

Lack of comparison with non-RPE model: As the authors have mentioned, there is a large community that disputes the possibility that DANs in insects actually represent RPEs, instead DANs may represent absolute reward values. However, there is no comparison or discussion of predictability or benefits of an RPE-based vs a reward-level based DAN mechanism.

We have modified the Discussion section “Previous mushroom body models”, now entitled “Other models of learning in the mushroom body”, so as to discuss the advantages of RPE coding over absolute reinforcement coding (stability and accuracy of reinforcement predictions), as well as to propose alternative mechanisms (synaptic weight decay and adaptation) that may replace RPE coding. Building and testing such mechanisms in a mushroom body model is well worth while, and something we are beginning to pursue, but we

feel it goes beyond the main focus of our current work, which is to investigate the plausibility of the RPE hypothesis as applied to the mushroom body.

SPECIFIC COMMENTS

In abstract and throughout RPs, RPEs, and RPTs are so similar, does the paper need all these abbreviations? It can be confusing when one reads RPs and expect RPEs

We have replaced our use of RPT (reinforcement per trial), using TAR (trial averaged reinforcement) instead. We have considered possible alternatives to RPs, but think this may still be the best acronym in order to explicitly connect RPs with RPEs, and to help make the text more concise given the number of times we refer to RPs.

Abstract: “[we] postulate as yet unobserved connectivity that overcomes limitations in the experimentally constrained model.” – This is vague and it not clear if what you postulate could be useful.

We have updated this sentence to indicate how the new connections are useful.

Line 34 & 39: blocking phenomenon has not been introduced. What is blocked? Why?

We now provide a brief description of the blocking phenomenon the first time it is mentioned in the Introduction, line 34.

Line 42: “Consistent with...” complex sentence and it is unclear what is supposed to be consistent with it’s role: organization or innervation. Rephrase.

We have moved the sentence in lines 61-62 forward to help clarify the mushroom body’s role in associative learning.

Line 47 & 51: protocerebral anterior medial (PAM) cluster; parentheses around abbreviation not the other way consistent with MBONs, DANs, etc.

Corrected.

Line 51: Cite figure 1c.

We have added this citation at the first mention of a model component, D+, on line 50.

Figure 1a Caption: “The red region highlights...” should say green.

Corrected.

Around eqn(2) – say that k is the firing rate of the KC population

Added on line 118.

Line 157: shouldn’t this be $r+m>0$ not $r>0$ or $m>0$?

Both versions are correct, as firing rates are rectified and cannot be negative. Thank you for the suggested alternative. Because it is more concise, we have adopted it.

Figure 2. I am sort of surprised the authors don't plot w^- and w^+ . As this paper concerns the mechanisms of learning, it would be interesting to see the changes in synaptic strength.

We appreciate that it is often helpful to see the weights explicitly. In our manuscript, however, very good proxies for the weights are the MBON firing rates. We originally chose to omit the weights so as to be more concise. We have now included the weights in Fig. 2 in order to show how closely the MBON rates reflect the synaptic weights, but have not plotted the weights in the other figures for brevity. We are happy to add the weights to the other figures as well if requested.

Figure 2b-d Caption: say explicitly in the caption that $\gamma = 1$.

Added.

Line 192: one more sentence on the consequences for learning. Or does it matter? Since in these two cases (when reinforcement exceeds upperbound), one of $M^{+/-}$ is zero and the correct action should certainly be taken.

We now provide an additional sentence to explain the consequences of the upper bound for learning. We do this after the first mention of the upper bound, on line 175.

Line 212: It's probably worth explaining this in a little more detail as it motivates Figures 3b-d. I found it difficult to understand why the model in 3a couldn't replicate appetitive vs aversive DANs. Really, the authors mean DANs should respond to the appearance (not removal) of reward. The inhibitory model in 3a only has DANs responding to the removal of a reward. Is this correct?

This is indeed correct for excitatory DAN responses, i.e. such that the DAN firing rate increases. In Fig. 3a, the appearance (removal) of a reward inhibits (excites) D^- , with no effect on D^+ . Similarly, the appearance (removal) of a punishment inhibits (excites) D^+ , with no effect on D^- . We have rephrased this sentence, using the Reviewer's suggested terminology, now on line 215.

Line 254: Explain why there are two solutions... essentially one comes from cost in (1) vs (4)?

We have rephrased the explanation regarding the two solutions to provide more clarity in the Results (now from line 255) and in the Methods (from line 730).

Line 269: "when $w^{+/-}$ changes reach zero", you don't show $w^{+/-}$. If the authors discuss an explicit value of w you need to provide plots of $w^{+/-}$ explicitly.

Please see our response above, about how MBON firing rates act as a proxy for the weights.

Figure 3e: The plots show an increase of the baseline firing rates (zero RPE firing rates) of the MBONs after each cycle of reward changes (compare trials [0-20], [80-100], and [160-180]). What is the mechanism of this? Are they unstable, i.e. will a continuation of the reward changes lead to a growth of the MBON baseline activity?

Thank you, this is important to clarify. The baseline increases only because $w^{+/-}$ are initialised with small values. The weights must therefore grow to begin with in order to establish accurate reinforcement predictions. Thereafter, the baseline remains constant, i.e. there is no instability. We now explain this on line 277 and in Supplemental Fig. 5.

Line 275: the authors might mention: blocking KC->DAN synaptic transmission (by setting $\gamma=0$).

Added, thank you for the suggestion.

Not at all clear which model is plotted (VSL vs MV) in Figure 4.

We now clarify in the Fig. 4 caption that the data is from the MV model, and that the VSlambda model produces almost identical behaviour.

Lines 299-303: Very precise statement with no data shown. Also, it is unclear why (even if it is in the Supp Fig 6 which I don't have) this would occur. More discussion (and figures) are necessary.

We have revised this section to better describe the results of our simulations. We have also provided two additional figures in the Supplementary Information: 1) show the probability of choosing highly rewarding cues as a function of the number of cues, and 2) the root mean squared error of RPs as a function of the number of cues.

Figure 5: $x=0$ and $y=0$ lines would be useful.

These have been added.

Figure 5: Slopes are small indicating that the model was more sensitive to perturbations than animals. This needs discussion.

We have added a sentence discussing this in line 363.

Lines 380 – 385: Again, “blocking” is used without a definition in the abstract and introduction. The precise meaning of this term might not be obvious for the wide audience of Nature comm (it wasn't for me). These lines should be at least given in the introduction. I wouldn't mention blocking in the abstract. Also, I would try to rephrase a little because X comes first (conditionally) then XY then Y. You might at least add “the subsequent association between Y and R...” to drive this sequential process home.

We have now provided a description of blocking in the Introduction (line 34), and have incorporated the Reviewer's suggestion in line 395.

Figure 6e. The x-axis labels are unclear. You might want to say $p_X = 0.8$; $p_Y=0.2$ with explanations in caption? Or independent vs non-independent? Instead of two very similar honeycombs.

Updated to independent and non-independent.

Prediction 2: Line 486: “If w_k^T cannot be determined, then any...”. What does “cannot be determined” mean? The authors seem to imply that it means that KCs have no effect on DANs (i.e. w_k^T doesn't affect the equation for plasticity), which seems like an extreme difference from the VSL model. What is really discussed here?

Here, we are referring to the difficulty with which KC → DAN inputs may be isolated in experiments. We have rephrased the section to make this more clear (now on line 501).

Prediction 2: It seems that adaptation of baseline firing rate to long-term reward signal could account for much of this limitation. Since this may solve a complicated connectivity issue (i.e. lend support to VSL over MV models), it seems this is worth exploring.

Please see our response to Reviewer 2's first General Comment above.

Line 565: It would be relevant to cite the Cassenear and Laurent papers from 2012 and 2007 in locusts.

We have added citations to these two papers in line 586.

Prediction 3: The valence of DANs is defined by its response to RPEs not reinforcement – This is, perhaps, the biggest conclusion of the work on the MV model. I would add text to highlight this in the results and potentially abstract. Perhaps also allude to other animal models where neurogenic amines are used for RPEs.

We have modified the text in Results, line 281, to further emphasise this point, and have included citations to the mammalian literature in the Discussion, line 509.

Prediction 5 and prediction 3 are connected: 5 is a consequence of 3.

Although these predictions both relate to the MV model, prediction 5 does not depend on prediction 3. Prediction 5 is specific to the plasticity rule expressed in Eq. 8, whereby synaptic plasticity in both w^+ and w^- requires information from both D^+ and D^- . Prediction 3, however, does not depend on whether Eq. 7 or Eq. 8 is used to specify the plasticity rule. We have clarified this in the section Discussion: Prediction 5, line 539.

Line 688: Shouldn't m_q have a w_m coefficient?

Thanks, corrected. We also added that $w_m = 1$ on line 715.

Line 704: More discussion is required (in the main text as well) as to why two solutions are presented. It is entirely unclear how (19) relates to (20) as it seems you've set $w_K = [1, 1, \dots, 1]$. Does "or" for eqn (19) mean there's an alternate derivation that is not shown or discussed? Is it supposed to be the same thing? (I highly doubt this). In any case, more discussion is necessary in the methods and only one (the one you actually use) should be presented in the main text.

We have attempted to clarify why two solutions are possible in the Results section (starting from line 255) and in the Methods (line 731). We presented both rules following a previous request from Reviewer 2 to outline why one rule is preferred over the other (we have provided a more thorough discussion of the flaws to one of the equations, Eq. 7, in the Supplementary Text). We think it helps to keep a presentation of both rules, as the flaws in one solution (which goes as $w_k * k - d_{-/+}$) provide reasons as to why the alternative solution (which goes as $d_{+/-} - d_{-/+}$) is preferable. This alternative solution underlies one of our major predictions that synapses should be modulated by DANs of both valences.

Reviewer #1 (Remarks to the Author):

The authors have satisfyingly addressed all points of concern raised by me. I recommend publication of the manuscript in its current form. I congratulate the authors to this informative work.

Reviewer #2 (Remarks to the Author):

In the Supp. Text, the authors now acknowledge the conflict between their model's prediction and the experimental results of Dylla et al.: that the model predicts that the DAN response to CS+ alone should be smaller after training than before, whereas Dylla et al actually saw increased responses to CS+ alone.

However, I don't understand their speculation on resolving the conflict. As I read the explanation, the authors seem to be conflating the conceptual distinction between CS and US with the actual distinction between the "r" and "w" terms in their equations. That is, they argue that because the Rescorla-Wagner rule doesn't affect how DANs respond to the CS+, we should interpret the decreased response to CS+ alone after training as representing a negative response to the omitted US superimposed on a normal response to the CS+ (lines 140-149, supp text). That is fine and helpful in conceptual terms, but it still doesn't change the fact that this negative response to the omitted US *comes from the decrease in w^+ * (and increase in w^- , in the MV model) (in the case of a punishment neuron d-). Thus, there is no reason to expect the decrease in w^+ to cause a decreased response to the US alone, because w^+ represents a synapse coming from Kenyon cells, which don't respond to the US. In contrast, there is every reason to expect the decrease in w^+ to cause a decreased response to the CS+ alone. I appreciate the conceptual distinction between "depressed response to CS" vs. "normal response to CS + negative response to omitted US", but I don't think I'm wrong to take the equation literally.

In addition, some of their specific points don't make sense. For example: "To compare 136 the behaviour of our model with Dylla et al., one would need to sum both the CS+ 137 response and the US response in the experiments."

That might be true, but the problem at stake is that the model predicts that the DAN response to CS+ alone should be smaller after training than before training. Dylla et al provide responses to the CS+ alone (i.e., no US, thus nothing to sum), both before and after training, so the data can be compared straightforwardly with the model. Thus, this point is irrelevant.

"Thus, when the punishment is later removed, and the CS+ is pre- 129 sented alone, D- will exhibit an inhibitory response."

Not necessarily an inhibitory response - merely a smaller excitatory response to the CS+ alone after training compared to the CS+ alone before training. (In the VSlambda model, there are literally no minus signs to produce inhibition in DANs: $d^- = r^- + w^+ * k + w_k * k$. In the MV model, there could be net inhibition, but there might simply be a reduced excitatory response to the CS+ alone after training.) The fact that the model doesn't necessarily predict inhibition per se also means that it's irrelevant that you can't measure inhibition with calcium imaging, as the conflict with data was never about inhibition, but rather the failure to observe a decrease in the response to the CS+ alone after training.

I don't see the conflict with Dylla et al as a dealbreaker for publication. As is often said, all models

are wrong but some are useful. If the authors can't come up with a more plausible explanation, I suggest that they just openly acknowledge the conflict and admit they can't explain it. I also suggest that they remove/replace lines 488-493, which I feel obfuscates the issue by addressing irrelevant factors like TD learning and ignoring the actual issue, which is the difference between experiments and the prediction of their model.

Reviewer #3 (Remarks to the Author):

The authors fully addressed my original comments.

We wish to thank the reviewers for their critical and constructive feedback. Please find our responses below in **bold text**.

Reviewer #1 (Remarks to the Author):

The authors have satisfyingly addressed all points of concern raised by me. I recommend publication of the manuscript in its current form. I congratulate the authors to this informative work.

Reviewer #2 (Remarks to the Author):

In the Supp. Text, the authors now acknowledge the conflict between their model's prediction and the experimental results of Dylla et al.: that the model predicts that the DAN response to CS+ alone should be smaller after training than before, whereas Dylla et al actually saw increased responses to CS+ alone.

However, I don't understand their speculation on resolving the conflict. As I read the explanation, the authors seem to be conflating the conceptual distinction between CS and US with the actual distinction between the "r" and "w" terms in their equations. That is, they argue that because the Rescorla-Wagner rule doesn't affect how DANs respond to the CS+, we should interpret the decreased response to CS+ alone after training as representing a negative response to the omitted US superimposed on a normal response to the CS+ (lines 140-149, supp text). That is fine and helpful in conceptual terms, but it still doesn't change the fact that this negative response to the omitted US *comes from the decrease in w^+ (and increase in w^- , in the MV model) (in the case of a punishment neuron d-). Thus, there is no reason to expect the decrease in w^+ to cause a decreased response to the US alone, because w^+ represents a synapse coming from Kenyon cells, which don't respond to the US. In contrast, there is every reason to expect the decrease in w^+ to cause a decreased response to the CS+ alone. I appreciate the conceptual distinction between "depressed response to CS" vs. "normal response to CS + negative response to omitted US", but I don't think I'm wrong to take the equation literally.

In addition, some of their specific points don't make sense. For example: "To compare 136 the behaviour of our model with Dylla et al., one would need to sum both the CS+ 137 response and the US response in the experiments."

That might be true, but the problem at stake is that the model predicts that the DAN response to CS+ alone should be smaller after training than before training. Dylla et al provide responses to the CS+ alone (i.e., no US, thus nothing to sum), both before and after training, so the data can be compared straightforwardly with the model. Thus, this point is irrelevant.

"Thus, when the punishment is later removed, and the CS+ is pre-
129 sented alone, D- will exhibit an inhibitory response."

Not necessarily an inhibitory response - merely a smaller excitatory response to the CS+ alone after training compared to the CS+ alone before training. (In the VSlambda model, there are literally no minus signs to produce inhibition in DANs: $d^- = r^- + w^+k + wk*k$. In the MV model, there could be net inhibition, but there might simply be a reduced excitatory response to the CS+ alone after training.) The fact that the model doesn't necessarily predict inhibition per se also means that it's irrelevant that you can't measure inhibition with calcium imaging, as the conflict with data was never about inhibition, but rather the failure to observe a decrease in the response to the CS+ alone after training.

I don't see the conflict with Dylla et al as a dealbreaker for publication. As is often said, all models

are wrong but some are useful. If the authors can't come up with a more plausible explanation, I suggest that they just openly acknowledge the conflict and admit they can't explain it. I also suggest that they remove/replace lines 488-493, which I feel obfuscates the issue by addressing irrelevant factors like TD learning and ignoring the actual issue, which is the difference between experiments and the prediction of their model.

We thank the reviewer for further clarifying their concern with our arguments. We agree that the reviewer's perspective on this matter – that the DAN response to the CS+ decreases, due to KC-MBON weight changes – is an important point that needs to be discussed. We have therefore emphasised the discrepancy between our results and the Dylla et al. experiments, and discussed this as a limitation of our model. To this end, we have made the following revisions. In lines 134-140 of the Supplementary Information, we describe the two possible interpretations of the isolated CS+ response, and state that the simplifications in our model cannot resolve them, and as such we cannot properly address the experimental data. In lines 142-147 of the Supplementary Information, we have limited our discussion of the TD learning rule, stating how it is known to produce the increased CS+ response seen in Dylla et al., but that aligning the theory with biologically plausible mushroom body models goes beyond the scope of our work. In the main text, lines 491-503, we now explicitly state the limitation of our model and how our results differ from the Dylla et al. data. We have also kept but updated our reference to TD learning in order to 1) address another reviewer's previous concerns, and 2) to emphasise that the Dylla et al. data need not conflict with the RPE hypothesis, but rather the limitations of our specific model.

Reviewer #3 (Remarks to the Author):

The authors fully addressed my original comments.